# Zap Q-learning with Nonlinear Function Approximation

**Shuhang Chen**
University of Florida
shuhangchen@ufl.edu

**Adithya M. Devraj**
Stanford University
adevraj@stanford.edu

**Fan Lu**
University of Florida
fan.lu@ufl.edu

**Ana Bušić**
INRIA - École Normale Supérieure
PSL Research University
ana.busic@inria.fr

**Sean P. Meyn**
University of Florida
meyn@ece.ufl.edu

## Abstract

Zap Q-learning is a recent class of reinforcement learning algorithms, motivated primarily as a means to accelerate convergence. Stability theory has been absent outside of two restrictive classes: the tabular setting, and optimal stopping. This paper introduces a new framework for analysis of a more general class of recursive algorithms known as stochastic approximation. Based on this general theory, it is shown that Zap Q-learning is consistent under a non-degeneracy assumption, even when the function approximation architecture is nonlinear. Zap Q-learning with neural network function approximation emerges as a special case, and is tested on examples from OpenAI Gym. Based on multiple experiments with a range of neural network sizes, it is found that the new algorithms converge quickly and are robust to choice of function approximation architecture.

## 1 Introduction

A primary goal of reinforcement learning (RL) is the creation of algorithms that are convergent, converge at the fastest possible rate, and result in a policy for control that has near optimal performance. This paper focuses on algorithm design to ensure stability of the algorithm, consistency, and techniques to obtain qualitative insight on the rate of convergence. One framework for algorithm design is the theory of stochastic approximation (SA). The main contribution of this work is a new class of Q-learning algorithms that are convergent even for nonlinear function approximation architectures, such as neural networks.

Consider a Markov decision process (MDP) model with state-input sequence $\{(X_n, U_n) : n \geq 0\}$, and let $\{Q^\theta(x, u) : \theta \in \mathbb{R}^d\}$ denote a family of approximations of the Q-function; the vector $\theta \in \mathbb{R}^d$ might correspond to weights in a neural network. One popular formulation of Q-learning is defined by the recursion,

$$\theta_{n+1} = \theta_n + \alpha_{n+1} \mathcal{D}_{n+1} \zeta_n \tag{1}$$

in which $\{\alpha_{n+1}\}$ is the non-negative step-size sequence, $\{\mathcal{D}_{n+1}\}$ is the scalar sequence of *temporal differences* [recalled in eq. (19a)], and $\{\zeta_n\}$ the *eligibility vectors*: a typical choice is

$$\zeta_n = \nabla_\theta Q^\theta(X_n, U_n)\Big|_{\theta=\theta_n} \tag{2}$$

The Q-learning algorithm of Watkins can be expressed as (1), with a linear function approximation $Q^\theta = \theta^\intercal \psi$, and the basis functions $\psi(x, u)$ being indicator functions of each state-input pair.

The theory of Q-learning with function approximation has not caught up with the famous success stories in applications. Consistency of the Q-learning algorithm in the tabular setting was established in the seminal work of Watkins and Dayan [50]. Counter-examples soon followed: the recursion (1) may fail to converge, even in the linear function approximation setting [3, 31]. Moreover, even when convergence holds, Q-learning can be extremely slow [45, 18, 16].

The so-called *ODE method* of SA theory is typically regarded as a method of analysis for stochastic recursions. We take the opposite view, regarding an ODE as a first step in algorithm design. This is motivated in part by the recent work [42, 39] concerning the value of high-resolution approximations of ODEs for applications in optimization, and the enormous insight gained from a careful inspection of candidate ODEs.

The Q-learning algorithms considered in the present work are designed to solve a root-finding problem of the form $\overline{f}(\theta^*) := \mathsf{E}[\zeta_n \mathcal{D}_{n+1}]\big|_{\theta=\theta^*} = 0$. For ODE design, we let $w_t \in \mathbb{R}^d$ denote the state of the ODE at time $t$, and seek a vector field $\nu\colon \mathbb{R}^{d+1} \to \mathbb{R}^d$ to define the evolution:

$$\tfrac{d}{dt} w_t = \nu(w_t, t)$$

The vector field is designed so that $w_t \to \theta^*$ from each initial condition, and so that the ODE solutions can be efficiently approximated using a discrete-time algorithm driven by observations.

One approach is to apply gradient descent to solve the non-convex optimization problem:

$$\min_\theta J(\theta) = \min_\theta \tfrac{1}{2}\overline{f}(\theta)^\intercal M \overline{f}(\theta), \qquad \text{with} \quad M > 0 \tag{3}$$

which results in the ODE with time homogeneous vector field:

$$\tfrac{d}{dt} w_t = -[\partial_\theta \overline{f}(w_t)]^\intercal M \overline{f}(w_t) \tag{4}$$

The GQ-learning algorithm of [31] can be regarded as a direct discrete-time translation of this ODE, using $M = \mathsf{E}[\zeta_n \zeta_n^\intercal]^{-1}$.

This approach is discussed in Nesterov's monograph [34, Section 4.4.1] for general root finding problems, who warns that it can lead to numerical instability: "...if our system of equations is linear, then such a transformation squares the condition number of the problem". He goes on to warn that it can lead to a "squaring the number of iterations" to obtain the desired error bound.

The main results of the present paper are related to the *Newton-Raphson flow*, defined by another time homogeneous vector field $\nu(w) = -[\partial_\theta \overline{f}(w)]^{-1}\overline{f}(w)$:

$$\tfrac{d}{dt} w_t = G_t \overline{f}(w_t), \qquad \text{with} \quad G_t = -[\partial_\theta \overline{f}(w_t)]^{-1} \tag{5}$$

A change of variables leads to the linear dynamics, $\tfrac{d}{dt}\overline{f}(w_t) = -\overline{f}(w_t)$, with solution

$$\overline{f}(w_t) = \overline{f}(w_0)\, e^{-t}, \qquad t \geq 0 \tag{6}$$

Thus, provided solutions to (5) are bounded, the algorithm is consistent in the sense that the limit points of $\{w_t\}$ lie in the set of roots $\Theta^* := \{\theta : \overline{f}(\theta) = 0\}$.

In most applications it is not possible to determine a-priori if the matrix $\partial_\theta \overline{f}(\theta)$ is full rank, which motivates a *regularized Newton-Raphson flow*:

$$\tfrac{d}{dt} w_t = -[\varepsilon I + A(w_t)^\intercal A(w_t)]^{-1} A(w_t)^\intercal \overline{f}(w_t), \qquad A(w_t) = \partial_\theta \overline{f}(w_t) \tag{7}$$

It is shown in Prop. A.6 that (7) is stable, provided $V = \|\overline{f}\|^2$ is a coercive function on $\mathbb{R}^d$; $V$ serves as a Lyapunov function for (7), giving

$$\lim_{t\to\infty} \overline{f}(w_t) = 0 \tag{8}$$

Hence the limit points of solutions lie in the set $\Theta^*$.

Details of the algorithms and the contributions of this paper require additional background. Consider the $d$-dimensional SA recursion of Robbins and Monro [38, 9]:

$$\theta_{n+1} = \theta_n + \alpha_{n+1} f(\theta_n, \Phi_{n+1}) \tag{9}$$

in which $\mathbf{\Phi}$ is an irreducible Markov chain on a finite state space $\mathsf{Z}$, $\{\alpha_n\}$ is a non-negative gain sequence, and $f\colon \mathbb{R}^d \times \mathsf{Z} \to \mathbb{R}^d$. It is assumed that $\mathbf{\Phi}$ has a unique invariant probability mass function (pmf) on $\mathsf{Z}$. The algorithm is designed to approximate roots of the function $\overline{f}(\theta) = \mathsf{E}[f(\theta, \Phi_{n+1})]$ (with expectation in steady-state). Under mild conditions, the SA recursion shares the same limit points as the ODE $\frac{d}{dt}w_t = \overline{f}(w_t)$ [30, 9, 6]. More recently it has been established that boundedness of the stochastic recursion follows from a stability condition for the ODE [10, 9, 37] (prior to this work, stability of the stochastic recursion required separate arguments [46]).

One approach to obtain a rate of convergence in SA is through the linearization:

$$\mathcal{E}_{n+1} = \mathcal{E}_n + \alpha_{n+1}[A_* \mathcal{E}_n + \Delta_{n+1}], \qquad \mathcal{E}_0 = \theta_0 - \theta^* \tag{10}$$

where $A_* = \partial_\theta \overline{f}(\theta^*)$ is called the *linearization matrix*. The sequence $\Delta_{n+1} := f(\theta^*, \Phi_{n+1})$ is assumed to admit a Central Limit Theorem (CLT) in the usual sense, with asymptotic covariance

$$\Sigma_\Delta = \sum_{k=-\infty}^{\infty} \mathsf{E}[\Delta_k \Delta_0^\mathsf{T}] \tag{11}$$

where the expectations are in steady state. The approximation $\mathcal{E}_n \approx \tilde{\theta}_n := \theta_n - \theta^*$ holds under additional stability assumptions on the stochastic recursion (9), which in particular leads to a CLT for the scaled error $\sqrt{n}\tilde{\theta}_n$ [30, 9, 6].

The asymptotic covariance $\Sigma_\theta$ in the CLT has a simple form, subject to the eigenvalue test:

$$\mathrm{Re}\,(\lambda) < -\tfrac{1}{2} \qquad \textit{for each eigenvalue } \lambda \textit{ of } A_* \tag{12}$$

Under this assumption, $\Sigma_\theta$ is the unique solution of the Lyapunov equation,

$$[\tfrac{1}{2}I + A_*]\Sigma_\theta + \Sigma_\theta[\tfrac{1}{2}I + A_*]^\mathsf{T} + \Sigma_\Delta = 0 \tag{13}$$

For a fixed but arbitrary initial condition $(\Phi_0, \mathcal{E}_0)$, denote $\Sigma_n = \mathsf{E}[\mathcal{E}_n \mathcal{E}_n^\mathsf{T}]$. The following bounds were obtained in [12] for the linear recursion (10):

(i)  If (12) holds, then $\Sigma_n = n^{-1}\Sigma_\theta + O(n^{-1-\delta})$ for some $\delta > 0$.

(ii)  If there exists an eigenvalue of $A_*$ with $\rho := -\mathrm{Re}\,(\lambda) < \tfrac{1}{2}$, and associated eigenvector $v$ satisfying $\Sigma_\Delta v \neq 0$, then the convergence rate of $\Sigma_n$ to zero is no faster than $n^{-2\rho}$.

Even though the recursion for Watkins' Q-learning is of the form (1), with $\mathcal{D}_{n+1}$ a *non-linear* function of $\theta_n$, techniques of [45] can be used to show that the estimates obtained using the non-linear recursion *couple* with the estimates of a linear recursion of the form (10). The slow convergence for Watkins' algorithm can then be explained by the fact that we are in case (ii) for the linearized recursion, whenever the discount factor satisfies $\gamma > \tfrac{1}{2}$: for a standard step-size rule, the maximal eigenvalue of $A_*$ in Watkins' Q-learning is $\lambda = -(1-\gamma)$, and the condition $\Sigma_\Delta v \neq 0$ holds under very mild conditions on the MDP [16]. It follows that the mean square error converges to zero at rate $n^{-2(1-\gamma)}$. For GQ-learning, it is shown in Appendix A.3 that the maximal eigenvalue is greater than $-(1-\gamma)^2$, which is consistent with Nesterov's warning.

The slow convergence can be remedied by scaling the step-size by a constant $g > 1$ (sufficiently large so that the matrix $\tfrac{1}{2}I + gA_*$ is Hurwitz). For tabular Q-learning any value satisfying $g > 1/[2(1-\gamma)]$ will suffice, while for GQ-learning the scaling must be increased beyond $1/[2(1-\gamma)^2]$. Unfortunately, this approach may lead to very high variance.

**Contributions**  **(i)** A generalization of the Zap SA algorithm of [16] is proposed.
**Zap SA Algorithm:** Initialize $\theta_0 \in \mathbb{R}^d$, $\widehat{A}_0 \in \mathbb{R}^{d\times d}$, $\varepsilon > 0$. Update for $n \geq 0$:

$$\widehat{A}_{n+1} = \widehat{A}_n + \beta_{n+1}\big[A_{n+1}(\theta_n) - \widehat{A}_n\big], \qquad A_{n+1}(\theta) := \partial_\theta f(\theta, \Phi_{n+1}) \tag{14a}$$

$$\theta_{n+1} = \theta_n + \alpha_{n+1}G_{n+1}f(\theta_n, \Phi_{n+1}), \qquad G_{n+1} := -[\varepsilon I + \widehat{A}_{n+1}^\mathsf{T}\widehat{A}_{n+1}]^{-1}\widehat{A}_{n+1}^\mathsf{T} \tag{14b}$$

The algorithm is designed so that it approximates the ODE (7), which requires $\alpha_n = o(\beta_n)$.

**(ii)** A special case of this new class of SA algorithms leads to a significant generalization of *Zap Q-learning*, for which convergence theory is obtained even in a *nonlinear* function approximation

setting. The reliability in neural network function approximation architectures is tested through simulations.

**(iii)** The main technical contribution of this paper is an extension of SA theory to Zap Q-learning, and as a byproduct also GQ-learning, by exploiting approximate convexity/concavity of the functions $f$ and $\bar{f}$ defined implicitly in (14).

Contribution (iii) resolves a significant challenge for both Zap Q-learning and GQ-learning: the *approximation* in stochastic approximation. Standard theory does not apply because $A(\theta) := \partial_\theta \bar{f}(\theta)$ is not continuous. An ODE approximation for GQ-learning is obtained in [31] through the assumption that noise $\{\Delta_n\}$ defined below (10) is martingale-difference. Assumption (Q3) of [16] is introduced to obtain an ODE approximation without this restrictive assumption on noise. However, this implicit assumption cannot be tested a-priori.

**Literature review**    The observation that many RL algorithms can be cast as SA first appeared in [46, 22]. Soon after, SA theory was applied to obtain stability theory for TD-learning with linear function approximation under minimal assumptions [48]; the authors discussed challenges for nonlinear approximation architectures.

In the case of Q-learning, ODE approximations are nonlinear and not understood outside of a few special cases (notably tabular, and optimal stopping with linear function approximation). There are many counterexamples showing that conditions on the function class are required in general, even in a linear function approximation setting [4] (also see [47, 43, 19]). There has also been progress for general linear function approximation: sufficient conditions for convergence of the basic Q-learning algorithm (1) was obtained in [32], with finite-$n$ bounds appearing recently in [13], and stability of GQ-learning was established in [31] subject to assumptions slightly stronger than (A1)–(A3) in the present paper. In particular, it is assumed in [31, Assumption L3] that $\partial_\theta \bar{f}(\theta)$ is everywhere nonsingular. In [23], the authors obtained regret bounds for Q-learning in an episodic setting, under a *linear MDP* (linear dynamics and linear rewards) assumption, stronger than the assumptions imposed here.

Stability theory for off-policy TD-learning faces similar challenges as Q-learning. A consistent algorithm is introduced in [44] for linear function approximation, using the same ideas as in [31]; this theory is extended to non-linear function approximation in [8].

To the best of our knowledge, the ODE (5) was introduced in the economics literature, which led to the comprehensive analysis by Smale [41] for smooth $\bar{f}$. The term *Newton-Raphson flow* for (5) was introduced in the deterministic control literature [40, 49]. The Zap SA algorithm was introduced at the same time, and based on the same ODE [15].

The motivation of [15] was centered entirely on optimizing the asymptotic covariance of stochastic approximation, and in particular Q-learning with tabular basis; see [30, 6] for history of convergence rate theory in SA, and [29, 28] for application to actor-critic methods. While the motivation here is stability, results in Section A.2 strongly suggest that the asymptotic covariance is approximately optimal for the regularized Zap Q-learning algorithm introduced here; a "tightness argument" is required to complete the proof.

The analysis in this paper can be cast in the general framework of stochastic approximation based on differential inclusions (see [9, Chapter 5] and its references). This general framework guided the research reported here. New in this paper is the proof of convergence of Zap Q-learning via an ODE approximation, made possible by the special structure of the recursion.

## 2    Zap Q-learning with Nonlinear Function Approximation

**Preliminaries**    We restrict to a discounted reward optimal control problem, with finite state space $\mathsf{X}$, finite input space $\mathsf{U}$, reward function $r : \mathsf{X} \times \mathsf{U} \to \mathbb{R}$, and discount factor $\gamma \in (0, 1)$. The Q-function is defined as the maximum over all possible input sequences $\{U_n : n \geq 1\}$ of the total discounted reward:

$$Q^*(x, u) := \max_{\boldsymbol{U}} \sum_{n=0}^{\infty} \gamma^n \mathsf{E}[r(X_n, U_n) \mid X_0 = x, U_0 = u], \qquad x \in \mathsf{X}, u \in \mathsf{U} \qquad (15)$$

Extensions to other criteria are straightforward (e.g., average cost or weighted shortest path).

Let $P_u$ denote the state transition matrix when input $u \in \mathsf{U}$ is taken. It is known that the Q-function is the unique solution to the Bellman equation [7]:

$$Q^*(x,u) = r(x,u) + \gamma \sum_{x' \in \mathsf{X}} P_u(x,x') \underline{Q}^*(x') \tag{16}$$

where $\underline{Q}(x) := \max_{u \in \mathsf{U}} Q(x,u)$ for any function $Q : \mathsf{X} \times \mathsf{U} \to \mathbb{R}$.

Consider a (possibly nonlinear) parameterized family of candidate approximations $\{Q^\theta : \theta \in \mathbb{R}^d\}$, wherein $Q^\theta : \mathsf{X} \times \mathsf{U} \to \mathbb{R}$ for each $\theta$, and the associated family of policies $\phi^\theta(x) \in \arg\max_u Q^\theta(x,u)$, $x \in \mathsf{X}$. To avoid ambiguities when the maximizer is not unique, we enumerate all stationary policies as $\{\phi^{(i)} : 1 \le i \le \ell_\phi\}$, and specify

$$\phi^\theta := \phi^{(\kappa)}, \qquad \text{where} \quad \kappa := \min\{i : \phi^{(i)}(x) \in \arg\max_u Q^\theta(x,u), \text{ for all } x \in \mathsf{X}\} \tag{17}$$

The recursion (1) is designed to compute an approximate solution of (16), defined as the solution to the root-finding problem:

$$\overline{f}(\theta^*) = 0, \qquad \text{with} \quad \overline{f}(\theta) := \mathsf{E}\big[\big(r(X_n, U_n) + \gamma \underline{Q}^\theta(X_{n+1}) - Q^\theta(X_n, U_n)\big)\zeta_n\big] \tag{18}$$

where the expectation is in steady state.

It is convenient to denote $\Phi_{n+1} := (X_{n+1}, X_n, U_{n+1}, U_n)$, with state space $\mathsf{Z} := \mathsf{X}^2 \times \mathsf{U}^2$. It is assumed throughout the paper that $\zeta_n = \zeta(\theta_n, \Phi_n)$ for a function $\zeta : \mathbb{R}^d \times \mathsf{Z} \to \mathbb{R}^d$.

**Algorithm**   The Zap SA algorithm (14) to solve (18) is obtained on specifying

$$\mathcal{D}(\theta_n, \Phi_{n+1}) := r(X_n, U_n) + \gamma \underline{Q}^{\theta_n}(X_{n+1}) - Q^{\theta_n}(X_n, U_n) \tag{19a}$$

$$f(\theta_n, \Phi_{n+1}) := \mathcal{D}(\theta_n, \Phi_{n+1})\zeta_n \tag{19b}$$

At points of differentiability, the derivative of $\overline{f}$ has a simple form:

$$A(\theta) := \partial_\theta \overline{f}(\theta) = \mathsf{E}[\zeta_n\big(\gamma \partial_\theta Q^\theta(X_{n+1}, \phi^\theta(X_{n+1})) - \partial_\theta Q^\theta(X_n, U_n)\big) + \mathcal{D}(\theta, \Phi_{n+1})\partial_\theta \zeta_n] \tag{20}$$

The Zap SA algorithm for Q-learning is exactly as described in (14) with $f$ defined in (19b), and $A_{n+1}(\theta)$ defined to be the term inside the expectation (20):

$$A_{n+1} = \zeta_n[\gamma \partial_\theta Q^{\theta_n}(X_{n+1}, \phi^{\theta_n}(X_{n+1})) - \partial_\theta Q^{\theta_n}(X_n, U_n)] + \mathcal{D}(\theta_n, \Phi_{n+1})\partial_\theta \zeta_n \tag{21a}$$

$$\widehat{A}_{n+1} = \widehat{A}_n + \beta_{n+1}\big[A_{n+1} - \widehat{A}_n\big] \tag{21b}$$

$$G_{n+1} = -[\varepsilon I + \widehat{A}_{n+1}^\mathsf{T} \widehat{A}_{n+1}]^{-1}\widehat{A}_{n+1}^\mathsf{T} \tag{21c}$$

$$\theta_{n+1} = \theta_n + \alpha_{n+1}G_{n+1}f(\theta_n, \Phi_{n+1}) \tag{21d}$$

Recall that $\phi^{\theta_n}$ is uniquely determined by (17) in the definition of $A_{n+1}$.

The step-size sequences $\{\alpha_n\}$ and $\{\beta_n\}$ satisfy standard requirements for two-time-scale SA algorithms [9]: $\beta_n/\alpha_n \to \infty$ as $n \to \infty$. For concreteness, in analysis we fix

$$\alpha_n = 1/(n + n_0), \quad \beta_n = \alpha_n^\rho, \qquad n \ge 1, \qquad \textit{with } n_0 \ge 1, \; \rho \in (0.5, 1) \tag{22}$$

The theory in this paper is focused on decreasing step-size mainly because theory of SA is more mature in this context. For constant step-size, with $\alpha_n \equiv \alpha$, $\beta_n \equiv \beta$, with $\beta = k\alpha$ for fixed $k \gg 1$, convergence of the algorithm can proceed by viewing the joint process $(\theta_n, \widehat{A}_n, \Phi_n)$ as a time-homogeneous Markov chain. Based on [10, Theorem 2.3] and [9, Chapter 9], it is conjectured that there exists $\bar{\alpha} > 0$ such that

$$\lim_{n \to \infty} \mathsf{E}[\|\theta_n - \theta^*\|^2] = O(\alpha), \qquad \alpha \in [0, \bar{\alpha}]. \tag{23}$$

Unfortunately, the mixing time of the Markov chain will increase with decreasing $\alpha$.

**Convergence Analysis** Given that $f$ in (19b) is non-smooth in $\theta$, analysis is cast in the theory of generalized subgradients of non-smooth functions. Consider first the temporal difference term $\mathcal{D} : \mathbb{R}^d \times \mathsf{Z} \to \mathbb{R}$. For each $z \in \mathsf{Z}$, the set of generalized subgradients of $\mathcal{D}(\theta, z)$ at $\theta_0$ is a convex set of row vectors, denoted by $\partial_\theta \mathcal{D}(\theta_0, z)$ [14, Chapter 10]. A vector $\vartheta \in \partial_\theta \mathcal{D}(\theta_0, z)$ has the defining property,

$$\vartheta v \leq \lim_{s \downarrow 0} \frac{\mathcal{D}(\theta_0 + sv, z) - \mathcal{D}(\theta_0, z)}{s}, \qquad v \in \mathbb{R}^d \tag{24}$$

The limit exists because $\mathcal{D}(\theta, z)$ is the pointwise maximum of smooth functions [14, Theorem 10.22]. The generalized subgradient of $f(\theta, z)$ exists under additional assumptions.

Recall that $\zeta_n = \zeta(\theta_n, \Phi_n)$ for each $n$. It is assumed henceforth that $\zeta$ is differentiable in $\theta$. In the presentation here we impose the additional assumption that the vector-valued function $\zeta$ has *non-negative entries*. We then obtain a version of the chain rule:

$$\partial_\theta f(\theta_0, z) = \{\zeta(\theta_0, z)\vartheta + \mathcal{D}(\theta_0, z)\partial_\theta \zeta(\theta_0, z) : \vartheta \in \partial_\theta \mathcal{D}(\theta_0, z)\} \tag{25}$$

We obtain in Lemma A.13 a similar representation for the set of generalized subgradients of $\overline{f}$:

$$\mathcal{A}(\theta) := \left\{ A \in \mathbb{R}^{d \times d} : Av \leq \lim_{s \downarrow 0} \frac{\overline{f}(\theta + sv) - \overline{f}(\theta)}{s}, \quad v \in \mathbb{R}^d \right\} \tag{26}$$

Non-negativity is relaxed in the supplementary material, based on a signed decomposition of $\zeta$.

**Assumptions:**

(A1) The joint process $(\boldsymbol{X}, \boldsymbol{U})$ is an irreducible Markov chain with unique invariant pmf $\varpi$.

(A2) $Q$ and $\zeta$ are Lipschitz continuous and twice continuously differentiable in $\theta$; $f(\theta, z)$ is Lipschitz continuous for each $z \in \mathsf{Z}$; $\|\overline{f}\|$ is coercive; $A^\intercal \overline{f}(\theta) \neq 0$ for $\theta \notin \Theta^*$, $A \in \mathcal{A}(\theta)$.

(A3) The set $\Theta^*$ is a singleton, so that there is a unique $\theta^* \in \mathbb{R}^d$ satisfying $\overline{f}(\theta^*) = 0$.

Assumption A1 rules out $\epsilon$-greedy policies and other parameter-dependent choices. It is likely that the theory can be extended using the general theory in [25, 9, Sections 6.2 and 6.3].

The second and third assumptions are first applied to the ODE (7): the coercive assumption in A2 implies boundedness of solutions, and this with A3 implies global asymptotic stability of (7).

It is also assumed throughout the paper that $\{\theta_n\}$ is bounded. This assumption is made to simplify the overall analysis and make the treatment of non-smooth $f$ accessible. In Section A.1 of the supplementary material, we provide additional sufficient conditions (which are easily satisfied for linear function approximation) to verify the boundedness assumption. We believe that (A1)-(A3) listed above suffice to establish boundedness of $\{\theta_n\}$ via an extension of the Borkar-Meyn theorem introduced in [10]. The following summarizes the main results of this paper:

**Theorem 2.1.** *Let $\{\theta_n\}$ be the parameter sequence obtained from the Zap Q-learning algorithm* (21), *with some fixed $\varepsilon > 0$. If this sequence is bounded, then*

(i) *If Assumptions A1–A2 hold, then $\lim_{n \to \infty} \overline{f}(\theta_n) = 0$ a.s..*

(ii) *If Assumptions A1–A3 hold, then $\lim_{n \to \infty} \theta_n = \theta^*$ a.s..*

$\square$

**Convergence Rate** Establishing a CLT for the scaled error sequence $\{\sqrt{n}\tilde{\theta}_n\}$ requires a "tightness bound" [9, Chapter 8, Lemma 5] and the following:

(A4) $f(\theta, z)$ is smooth in a neighborhood of $\theta^*$ for $z \in \mathsf{Z}$, and $A(\theta^*) = \partial_\theta \overline{f}(\theta^*)$ is non-singular.

Tightness is used to justify an approximation of the algorithm with its linearization (10). The proof of tightness is left to future work. In Section A.2 of the supplementary material we consider the linearization, and show that the asymptotic covariance satisfies $\Sigma_\theta = \Sigma_\theta^* + \varepsilon^2 \Sigma_\theta^{(2)} + O(\varepsilon^3)$, where $\Sigma_\theta$ is the asymptotic covariance obtained for Zap Q-learning (21), $\Sigma_\theta^*$ is the optimal covariance, and $\Sigma_\theta^{(2)}$ is identified in Prop. A.3.

**Overview of Proof of Thm. 2.1**   [complete proofs are found in the supplementary material]

The first step is analysis of the ODE (7) that $\{\theta_n\}$ aims to approximate. It is shown in Prop. A.6 that the ODE (7) admits at least one solution $\{w_t : t \geq 0\}$ from each initial condition; this is non-trivial, since the right hand side is discontinuous. We then conclude under (A2) that $\lim_{t \to \infty} \overline{f}(w_t) = 0$ for any solution. If in addition (A3) holds, then the ODE is globally asymptotically stable. These conclusions are obtained through a uniform approximation based on a family of smooth vector fields.

Discontinuity of the ODE and the stochastic recursion presents a greater challenge when we turn to establishing solidarity between the ODE and its stochastic counterpart. The main idea in this part of the analysis is most easily described for a special case: a linear parameterization $Q^\theta = \psi^\mathsf{T}\theta$, and non-negativity of $\zeta_n$ so that the chain rule (25) holds. We then obtain subgradients of the components of $f$, which implies the following component-wise bounds:

$$
\begin{aligned}
f(\theta_n + v, \Phi_{n+1}) &= \max_u \left\{ r(X_n, U_n) + [\gamma\psi(X_{n+1}, u) - \psi(X_n, U_n)]^\mathsf{T}(\theta_n + v) \right\}\zeta_n \\
&\geq f(\theta_n, \Phi_{n+1}) + A_{n+1}v, \qquad v \in \mathbb{R}^d
\end{aligned}
\tag{27}
$$

The update equation for $\widehat{A}_{n+1}$ in (21b) is used to obtain the averaged version of (27):

$$
\overline{f}(\theta_n + v) \geq \overline{f}(\theta_n) + \widehat{A}_{n+1}v + o(1), \qquad v \in \mathbb{R}^d,\ \|v\| \leq 1
\tag{28}
$$

where $o(1) \to 0$ as $n \to \infty$, uniformly in $v$. This implies that $\widehat{A}_{n+1}$ is close to the set of the subgradients $\mathcal{A}(\theta_n)$, in the sense made precise in Prop. A.18.

The arguments are considerably more complex when $Q^\theta$ is non-linear, and the positivity assumption on $\zeta_n$ is relaxed. In particular, without positivity, neither $f$ nor $\overline{f}$ admit the generalized subgradients. Fortunately, the techniques developed for the special case can be adapted to demonstrate that $\widehat{A}_{n+1}v$ approximates the directional derivative $\overline{f}'(\theta_n; v)$ for each $v$ and all large $n$.

Once all of these technical results are established, we obtain solidarity between Zap algorithm (21) and the ODE (7) in the sense that $\lim_{n\to\infty} \overline{f}(\theta_n) = \lim_{t\to\infty} \overline{f}(w_t) = 0$, along with the other conclusions of Thm. 2.1.

## 3   Numerical Results

The Zap Q-learning algorithm was tested on three examples from OpenAI gym: Mountain Car, Acrobot, and Cartpole [1]. The approximation of $Q^\theta$ was obtained based on a neural network, so that the parameter $\theta \in \mathbb{R}^d$ represents weights in the neural network. Rather than achieving the best score for specific tasks, the objective of the experiments surveyed in this section was to investigate the stability and consistency of the Zap Q-learning algorithm across different domains, and varying neural network sizes. Common in each experiment: a feedforward neural network that is fully connected, using the Leaky ReLU activation function.

The goal in each of the three examples is to collect as many rewards as possible before the state reaches a terminal set denoted $\mathcal{S} \subset \mathsf{X}$. To avoid infinite values we introduce a deterministic upper bound $\overline{\tau} \geq 1$, and consider the bounded horizon $\tau = \min(\overline{\tau}, \tau_\mathcal{S})$ with $\tau_\mathcal{S} = \min\{n \geq 1 : X_n \in \mathcal{S}\}$. The Q-function is denoted

$$
Q^*(x, u) := \max \mathsf{E}\Big[\sum_{n=0}^{\tau-1} r(X_n, U_n) \mid X_0 = x,\, U_0 = u\Big],
$$

which for $\overline{\tau} = \infty$ solves the Bellman equation

$$
Q^*(x, u) = r(x, u) + \mathsf{E}[\mathbb{I}\{X_1 \notin \mathcal{S}\}\underline{Q}^*(X_1) \mid X_0 = x, U_0 = u]
\tag{29}
$$

Following the roadmap outlined in Section 2, we seek an approximate solution to (29) based on a root-finding problem analogous to (18): find $\theta^*$ such that

$$
\overline{f}(\theta^*) = 0, \quad \text{with} \quad \overline{f}(\theta) := \mathsf{E}\big[\zeta_n\big(r(X_n, U_n) + \mathbb{I}\{X_{n+1} \notin \mathcal{S}\}\underline{Q}^\theta(X_{n+1}) - Q^\theta(X_n, U_n)\big)\big]
\tag{30}
$$

where the distribution of $X_0$ is given. The Zap-Q algorithm (21) is easily adapted to the modified definition of $\overline{f}$ in (30).

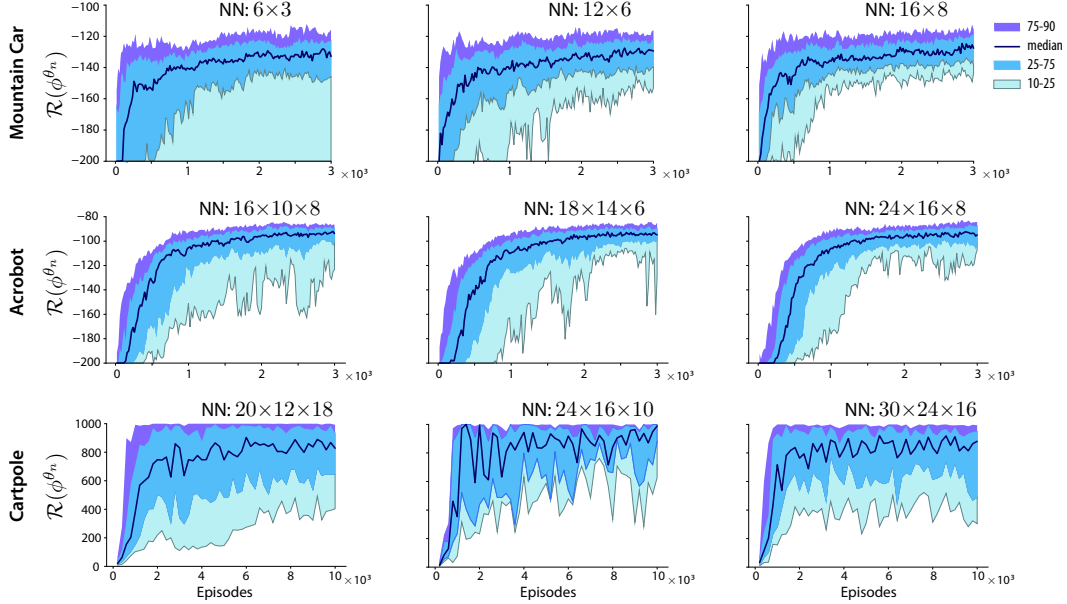

Figure 1: Average rewards for the three examples, shown by percentile.

The performance of the greedy policy $\phi^\theta$ induced by $Q^\theta$ is denoted

$$\mathcal{R}(\phi^\theta) := \mathsf{E}\Big[ \sum_{n=0}^{\tau-1} r(X_n, \phi^\theta(X_n)) \Big] \tag{31}$$

Specifics of the meta-parameters and the details of how (31) was estimated are contained in Section A.6.

Two minor modifications of the algorithm were used in these experiments to reduce complexity:
*1. Periodic gain update.* An integer $N_d > 1$ was fixed, and the gain $G_n$ appearing in (21d) was updated only for integer multiples of $N_d$. In particular, the matrix inversion step was only performed at these iterations. Letting $N$ denote the total number of iterations, the overall complexity of running this algorithm for $N$ iterations is in the worst-case $O(Nd^3/N_d + Nd^2)$ (see Section A.6 of the supplementary material for further discussion on complexity). We observed that $N_d = 50$ worked well for all experiments, and the performance was unchanged from $N_d = 1$.

*2. Periodic eligibility update.* The definition of the eligibility vector in (2) was modified to break $\theta$-dependency: $\zeta_n := \nabla_\theta Q^{\theta_{i(n)}}(X_n, U_n)$, with $i(n) := (\lfloor n/N_\zeta + 1 \rfloor - 1)N_\zeta$, and $N_\zeta = 2000$. We then ignored the term $\mathcal{D}(\theta_n, \Phi_{n+1})\partial_\theta \zeta_n$ when computing $A_{n+1}$ based on (21a). For comparison, we performed experiments in which this term was included (increasing complexity considerably since second derivatives are required [11]), and saw no improvement in performance.

Experiments were performed with both decreasing step-size (defined in (22)), and constant step-size. We found that constant step-size implementations were more reliable for the Mountain car and Acrobot examples, and the diminishing step-size gave better results for Cartpole. Figure 1 shows results obtained for these choices. The size of neural networks indicated in the figure refers only to hidden layers. To obtain the quantiles shown, each experiment was repeated 50 times, with parameters randomly initialized by the Kaiming uniform method [20, 36]. The first column shows results from the smallest network for which we obtained reliable results for the particular example.

## 4 Conclusions

Zap Q-learning is provably consistent with nonlinear function approximation, under very general conditions. Theoretical questions remain, such as extension to more general exploration strategies, and convergence properties for more general step-size rules. There are also architectural questions. For example, the definition of the eligibility vector (2) is not sacred. Better overall performance, and

simpler conditions for convergence may be achieved through alternatives (there are obvious choices for deterministic control systems rather than MDPs).

Better algorithm design requires a better theory for function approximation in Q-learning. The parameter estimates in both Zap Q-learning and the basic algorithm (1) are solutions to the root-finding problem (18). How do we bound the Bellman error, or the absolute error $|Q^{\theta^*} - Q^*|$? The goal is to create a theory as complete as TD-learning with linear function approximation, for which vector space concepts bring crisp answers [48, Theorem 1].

The matrix-inversion step in the algorithm may be a barrier to application of Zap-Q in some problems. A simple approach to reduce complexity is described in the numerical results, and we expect to obtain much more efficient implementations, perhaps by applying a distributed implementation [2], and adapting techniques from stochastic optimization.

We are currently exploring the application of the techniques in this paper to analyze Deep Q-learning [33], and application of Zap-SA to actor-critic methods.

**Acknowledgments:** Financial support from ARO grant W911NF1810334 is gratefully acknowledged. Additional support from EPCN 1935389 & CPS 1646229, and French National Research Agency grant ANR-16-CE05-0008.

## Broader Impact

This paper has focused on formulating Q-learning algorithms for which reliability is guaranteed by design. It is hoped that it will inspire the creation of a larger tool kit for ODE algorithm design, and methods to more efficiently translate the ODE to obtain better algorithms for reinforcement learning, especially in other contexts such as actor-critic methods. Given the importance of stochastic approximation in so many other fields, such as optimization, it is hoped that the impact will extend far beyond reinforcement learning.

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
