[Supplementary Material]

# A Appendices

Figure 2: Organization of appendices.

Figure 2 contains an overview of the appendices to help the reader navigate through the technical results leading to the proof of the main results, summarized in Thm. 2.1.

Sections A.1–A.3 collect proofs of miscellaneous technical claims: Section A.1 contains sufficient conditions to ensure boundedness of $\{\theta_n\}$. Section A.2 is focused on the linearization of Zap SA algorithm (14), and based on this a formula for the asymptotic covariance of Zap SA is obtained. The proof that the maximal eigenvalue for GQ-learning with tabular basis satisfies $\text{Real}(\lambda(A_*)) \geq -(1-\gamma)^2$ is contained in Section A.3.

Analysis of the Zap ODE (7) is contained in Section A.4, including existence of solutions, and the consistency result $\lim_{t \to \infty} \overline{f}(w_t) = 0$. Section A.5 finishes the proof of Thm. 2.1 by establishing solidarity between the ODE and the stochastic recursion. It is simplest to begin in the special case in which $\zeta$ has non-negative entries, which is the focus of Sections A.5.2 and A.5.3. The arguments are extended in Section A.5.4, where the non-negativity is relaxed.

## A.1 Establishing boundedness of the parameter estimates

Suppose that the following limits exist:

$$Q_\infty^\theta(x, u) = \lim_{m \to \infty} m^{-1} Q^{m\theta}(x, u), \qquad x \in \mathsf{X}, \ u \in \mathsf{U}$$

$$\zeta_\infty(\theta, z) = \lim_{m \to \infty} \zeta(m\theta, z), \qquad z \in \mathsf{Z}$$

where the limiting functions are twice continuously differentiable. The global Lipschitz conditions in (A2) imply that the gradients also converge, and the convergence is uniform on compact sets. We then obtain a vector field for the "ODE at infinity" introduced in [10]:

$$\overline{f}_\infty(\theta) := \lim_{m \to \infty} m^{-1} \overline{f}(m\theta) = \mathsf{E}\big[(\gamma \underline{Q}_\infty^\theta(X_{n+1}) - Q_\infty^\theta(X_n, U_n))\zeta_\infty(\theta, \Phi_n)\big]$$

and a similar definition for $f_\infty(\theta, z)$. The associated regularized Newton-Raphson flow "at infinity" is similar to (7):

$$\frac{d}{dt} w_t = -[\varepsilon I + A_\infty(w_t)^\intercal A_\infty(w_t)]^{-1} A_\infty(w_t)^\intercal \overline{f}_\infty(w_t), \qquad A_\infty(w_t) = \partial_\theta \overline{f}_\infty(w_t) \qquad (32)$$

With $\mathcal{A}_\infty(\theta)$ defined as in (26) with respect to $\overline{f}_\infty$, assume the following:

(A2$_\infty$) The functions $Q_\infty$ and $\zeta_\infty$ are Lipschitz continuous and twice continuously differentiable in $\theta$ in any open set not containing the origin; $f_\infty(\theta, z)$ is Lipschitz continuous for each $z \in \mathsf{Z}$; $A^\intercal \overline{f}_\infty(\theta) \neq 0$ for all $\theta \neq 0$ and $A \in \mathcal{A}_\infty(\theta)$.

Assumptions (A2) and (A2$_\infty$) are identical when the function approximation $Q^\theta$ is linear, and $\zeta = \nabla Q^\theta$.

The function $\|\overline{f}_\infty\|$ is coercive under (A2$_\infty$) since $\overline{f}_\infty$ is radially linear: $\overline{f}_\infty(m\theta) = m\overline{f}_\infty(\theta)$ for any $\theta$ and any $m \geq 0$. Prop. A.6 can be adapted to show that (32) is globally asymptotically stable. [9, Sections 6.3, Theorem 9] explains how stability of the ODE implies stability of the SA algorithm.

## A.2 Asymptotic covariance of regularized Zap SA

We first introduce a standard result in linear system theory [24, Theorem 2.6-1].

**Lemma A.1.** *If $A \in \mathbb{R}^{d \times d}$ is Hurwitz and $\Sigma_\Delta \in \mathbb{R}^{d \times d}$ is positive semi-definite, then there exists a unique solution $\Sigma \geq 0$ that solves the Lyapunov equation,*

$$A\Sigma + \Sigma A^\mathsf{T} + \Sigma_\Delta = 0\,,$$

*whose solution can be expressed*

$$\Sigma = \int_0^\infty \exp(A\tau)\Sigma_\Delta \exp(A^\mathsf{T}\tau)\, d\tau$$

Let $\{\mathcal{E}_n^G\}$ be the sequence obtained by the stochastic linear recursion (10) with matrix gain $G \in \mathbb{R}^{d \times d}$:

$$\mathcal{E}_{n+1}^G = \mathcal{E}_n^G + \alpha_{n+1} G[A_* \mathcal{E}_n^G + \Delta_{n+1}]\,, \qquad \mathcal{E}_0^G = \theta_0 - \theta^* \tag{33}$$

Denote the asymptotic covariance of $\{\mathcal{E}_n^G\}$ by $\Sigma_\theta^G := \lim_{n \to \infty} n\mathsf{E}[\mathcal{E}_n^G (\mathcal{E}_n^G)^\mathsf{T}]$. According to the eigenvalue test (12), $\Sigma_\theta^G$ is finite if $\frac{1}{2}I + GA_*$ is Hurwitz. It is well known that the matrix gain $G_* = -A_*^{-1}$ achieves the minimal asymptotic covariance:

$$\Sigma_\theta^* = A_*^{-1}\Sigma_\Delta (A_*^{-1})^\mathsf{T} \tag{34}$$

The following result is standard in stochastic approximation [5, Part I, Section 3.2.3, Proposition 4], and quantifies optimality of $\Sigma_\theta^*$.

**Lemma A.2.** *Suppose that $A_*$ and $\frac{1}{2}I + GA_*$ are Hurwitz. Then,*

(i) *The asymptotic covariance $\Sigma_\theta^G \geq 0$ uniquely solves the Lyapunov equation:*

$$(\tfrac{1}{2}I + GA_*)\Sigma_\theta^G + \Sigma_\theta^G(\tfrac{1}{2}I + GA_*)^\mathsf{T} + G\Sigma_\Delta G^\mathsf{T} = 0$$

(ii) *The sub-optimality gap $\widetilde{\Sigma}_\theta^G = \Sigma_\theta^G - \Sigma_\theta^* \geq 0$ uniquely solves the Lyapunov equation:*

$$(\tfrac{1}{2}I + GA_*)\widetilde{\Sigma}_\theta^G + \widetilde{\Sigma}_\theta^G(\tfrac{1}{2}I + GA_*)^\mathsf{T} + (G + A_*^{-1})\Sigma_\Delta (G + A_*^{-1})^\mathsf{T} = 0 \tag{35}$$

For any symmetric matrix $S \in \mathbb{R}^{d \times d}$, denote by $\lambda(S)$ the set of its eigenvalues.

**Proposition A.3.** *Suppose $A_* \in \mathbb{R}^{d \times d}$ is Hurwitz, and denote $G_\varepsilon = -[\varepsilon I + A_*^\mathsf{T} A_*]^{-1} A_*^\mathsf{T}$. If $0 < \varepsilon < \lambda_{\min}(A_*^\mathsf{T} A_*)$, then $\frac{1}{2}I + G_\varepsilon A_*$ is Hurwitz, so that the matrix gain $G_\varepsilon$ in the linear recursion (33) results in a finite asymptotic covariance $\Sigma_\theta^\varepsilon$. Moreover, the follow approximation holds:*

$$\Sigma_\theta^\varepsilon = \Sigma_\theta^* + \varepsilon^2 \Sigma_\theta^{(2)} + O(\varepsilon^3)\,, \qquad with\ \Sigma_\theta^{(2)} = (A_* A_*^\mathsf{T} A_*)^{-1}\Sigma_\Delta (A_*^\mathsf{T} A_* A_*^\mathsf{T})^{-1}. \tag{36}$$

*Proof.* The set of eigenvalues of $\frac{1}{2}I + G_\varepsilon A_*$ admits the following representations:

$$\begin{aligned}
\lambda(\tfrac{1}{2}I + G_\varepsilon A_*) &= \{\tfrac{1}{2} - \lambda : \lambda \in \lambda([\varepsilon I + A_*^\mathsf{T} A_*]^{-1} A_*^\mathsf{T} A)\} \\
&= \{\tfrac{1}{2} - \tfrac{1}{\lambda} : \lambda \in \lambda(\varepsilon (A_*^\mathsf{T} A_*)^{-1} + I)\} \\
&= \{\tfrac{1}{2} - \tfrac{1}{\varepsilon\lambda + 1} : \lambda \in \lambda((A_*^\mathsf{T} A_*)^{-1})\} \\
&= \{\tfrac{1}{2} - \tfrac{1}{\varepsilon/\lambda + 1} : \lambda \in \lambda(A_*^\mathsf{T} A_*)\}
\end{aligned}$$

Given $0 < \varepsilon < \lambda_{\min}(A_*^\mathsf{T} A_*)$, the eigenvalues of $\frac{1}{2}I + G_\varepsilon A_*$ are real and strictly negative. In particular, this matrix is Hurwitz, as claimed.

We next establish the approximation (36). By Lemma A.2 (iii), $\widetilde{\Sigma}_\theta^\varepsilon = \Sigma_\theta^\varepsilon - \Sigma_\theta^*$ solves the Lyapunov equation (35) with $G$ replaced by $G_\varepsilon$. Denoting $\widetilde{G}_\varepsilon = G_\varepsilon + A_*^{-1}$, we obtain by Lemma A.1,

$$\widetilde{\Sigma}_\theta^\varepsilon = \int_0^\infty \exp([\tfrac{1}{2}I + G_\varepsilon A_*]\tau)\widetilde{G}_\varepsilon \Sigma_\Delta \widetilde{G}_\varepsilon^\mathsf{T} \exp([\tfrac{1}{2}I + G_\varepsilon A_*]^\mathsf{T}\tau)\, d\tau \tag{37}$$

A Taylor series representation of matrix inverse results in the following:

$$G_\varepsilon A_* = -[\varepsilon I + A_*^\intercal A_*]^{-1} A_*^\intercal A_* = -[I + \varepsilon(A_*^\intercal A_*)^{-1}]^{-1} = -[I - \varepsilon A_*^{-1}(A_*^\intercal)^{-1}] + O(\varepsilon^2)$$

$$\widetilde{G}_\varepsilon = G_\varepsilon + A_*^{-1} = [G_\varepsilon A_* + I]A_*^{-1} = \varepsilon A_*^{-1}(A_*^\intercal)^{-1}A_*^{-1} + O(\varepsilon^2)$$

With $\Sigma_\theta^{(2)} = (A_* A_*^\intercal A_*)^{-1}\Sigma_\Delta(A_*^\intercal A_* A_*^\intercal)^{-1}$, the integral in (37) becomes

$$\widetilde{\Sigma}_\theta^\varepsilon = \varepsilon^2 \int_0^\infty \exp(-[\tfrac{1}{2}I - \varepsilon A_*^{-1}(A_*^\intercal)^{-1}]\tau)\Sigma_\theta^{(2)} \exp(-[\tfrac{1}{2}I - \varepsilon A_*^{-1}(A_*^\intercal)^{-1}]^\intercal \tau)d\tau + O(\varepsilon^3) \quad (38)$$

Another Taylor series expansion for matrix exponential gives

$$\exp(-[\tfrac{1}{2}I - \varepsilon A_*^{-1}(A_*^\intercal)^{-1}]\tau) = \exp(-\tfrac{\tau}{2}I) + \varepsilon\tau A_*^{-1}(A_*^\intercal)^{-1}\exp(-\tfrac{\tau}{2}I) + O(\varepsilon^2)$$

Consequently, the integral in (38) can be rewritten as

$$\widetilde{\Sigma}_\theta^\varepsilon = \varepsilon^2 \int_0^\infty \exp(-\tfrac{\tau}{2}I)\Sigma_\theta^{(2)}\exp(-\tfrac{\tau}{2}I)d\tau + O(\varepsilon^3)$$

$$= \varepsilon^2 \Sigma_\theta^{(2)} + O(\varepsilon^3)$$

$\square$

### A.3 Eigenvalue test for GQ-learning

Consider the linear function approximation architecture: $Q^\theta(x,u) = \psi(x,u)^\intercal\theta$, where $\psi : \mathsf{X} \times \mathsf{U} \to \mathbb{R}^d$ is the basis function. With eligibility vector $\zeta_n := \psi(X_n, U_n)$, let $\overline{f}$ be defined by (18). GQ-learning [31] aims to solve the root finding problem (18), transformed into the non-convex optimization problem (3).

The GQ-learning [31] algorithm is the two-time scale SA algorithm,

$$\theta_{n+1} = \theta_n + \alpha_{n+1}[\mathcal{D}(\theta_n, \Phi_{n+1})\zeta_n - \gamma\varphi_{n+1}^\intercal\zeta_n\psi(X_{n+1}, \phi^{\theta_n}(X_{n+1}))] \quad (39a)$$

$$\varphi_{n+1} = \varphi_n + \beta_{n+1}\zeta_n[\mathcal{D}(\theta_n, \Phi_{n+1}) - \psi(X_n, U_n)^\intercal\varphi_n] \quad (39b)$$

where $\{\beta_n\}$ and $\{\alpha_n\}$ are non-negative step-size sequences satisfying $\alpha_n/\beta_n \to 0$ as $n \to \infty$. The fast time scale recursion (39b) for $\{\varphi_n\}$ is designed so that $\varphi_n \approx M\overline{f}(\theta_n)$ for large $n$. It follows that the ODE approximation of (39a) is (4).

**Proposition A.4.** *The linearization matrix for GQ-learning at $\theta^*$ is given by $A_{GQ} = -A_*^\intercal M A_*$, whenever $\theta^*$ is a solution to $A(\theta)^\intercal M\overline{f}(\theta) = 0$. With the tabular basis: $\psi_k(x,u) = \mathbb{I}\{x = x^k, u = u^k\}, 1 \le k \le \ell_x \cdot \ell_u$, there is an eigenvalue $\lambda_{GQ}$ of $A_{GQ}$ satisfying*

$$\lambda_{GQ} \ge -(1-\gamma)^2$$

We first introduce some notation for tabular Q-learning. For any deterministic stationary policy $\phi : \mathsf{X} \to \mathsf{U}$, let $S_\phi$ denote the substitution operator, defined for any function $Q : \mathsf{X} \times \mathsf{U} \to \mathbb{R}$ by $S_\phi Q(x) = Q(x, \phi(x))$. With $P$ viewed as a matrix with $\ell_x \cdot \ell_u$ rows and $\ell_x$ columns, $PS_\phi$ can be interpreted as the transition matrix for the joint process $(\boldsymbol{X}, \boldsymbol{U})$ when $\boldsymbol{U}$ is defined using policy $\phi$ [15]. Then $\overline{f}(\theta)$ can be written in matrix form

$$\overline{f}(\theta) = \Pi r + \Pi[\gamma PS_{\phi^\theta} - I]\theta \quad (40)$$

where $\Pi$ is a diagonal matrix with entries: $\Pi(k,k) := \varpi(x^k, u^k)$ and $r$ is a vector with entries: $r(k) := r(x^k, u^k)$. The derivative $A(\theta)$ of $\overline{f}(\theta)$ is given by

$$A(\theta) = \Pi[\gamma PS_{\phi^\theta} - I]$$

*Proof.* The matrix $A_{\mathrm{GQ}}$ is the derivative of $-A(w_t)^\intercal M\overline{f}(w_t)$ at $\theta^*$. For the tabular case, by (40),

$$A_{\mathrm{GQ}} = -[\gamma PS_{\phi^*} - I]^\intercal \Pi[\gamma PS_{\phi^*} - I] = -H^\intercal H,$$

with $H := \Pi^{1/2}[I - \gamma PS_{\phi^*}]$. It suffices to show that $H^\intercal H$ has a positive eigenvalue less than $(1-\gamma)^2$.

Since $H^{-1}$ is a positive and irreducible matrix, we can apply the same arguments as in [15, Theorem 3.3] to bound the Perron-Frobenius eigenvalue as follows:

$$\lambda_{\mathrm{PF}} \geq \frac{1}{1-\gamma} \min_{x,u} \frac{1}{\sqrt{\varpi(x,u)}}$$

Therefore, $H$ has positive eigenvalue $\lambda_H = \lambda_{\mathrm{PF}}^{-1}$ such that

$$\lambda_H \leq (1-\gamma) \max_{x,u} \sqrt{\varpi(x,u)}$$

Applying [21, Theorem 5.6.9] we obtain the complementary bound

$$\lambda_H \geq \sigma_{\min}(H) = \sqrt{\lambda_{\min}(H^{\mathsf{T}}H)}$$

and combining the two implies:

$$\lambda_{\min}(H^{\mathsf{T}}H) \leq \lambda_H^2 \leq (1-\gamma)^2 \big(\max_{x,u} \sqrt{\varpi(x,u)}\big)^2 \leq (1-\gamma)^2$$

$\square$

## A.4  ODE Analysis

To obtain the existence of a solution to (7), we first consider an ideal smooth setting:

**Proposition A.5.** *Consider the following conditions for the function $\overline{f}$:*

(a) *$\overline{f}$ is globally Lipschitz continuous and continuously differentiable. Hence $A(\cdot)$ is a bounded matrix-valued function.*

(b) *$\|\overline{f}\|$ is coercive. That is, $\{\theta : \|\overline{f}(\theta)\| \leq n\}$ is compact for each $n$.*

(c) *The function $\overline{f}$ has a unique zero $\theta^*$, and $A^{\mathsf{T}}(\theta)\overline{f}(\theta) \neq 0$ for $\theta \neq \theta^*$. Moreover, the matrix $A_* = A(\theta^*)$ is non-singular.*

*The following hold for solutions to the ODE (7) under increasingly stronger assumptions:*

(i) *If (a) holds then for each $t$, and each initial condition*

$$\tfrac{d}{dt}\overline{f}(w_t) = -A(w_t)[\varepsilon I + A(w_t)^{\mathsf{T}}A(w_t)]^{-1} A(w_t)^{\mathsf{T}}\overline{f}(w_t) \tag{41}$$

(ii) *If in addition (b) holds, then the solutions to the ODE are bounded, and*

$$\lim_{t \to \infty} A(w_t)^{\mathsf{T}}\overline{f}(w_t) = 0 \tag{42}$$

(iii) *If (a)–(c) hold, then (7) is globally asymptotically stable.* $\square$

*Proof.* The result (i) follows from the chain rule and the definitions.

The proof of (ii) is based on the Lyapunov function $V(w) = \frac{1}{2}\|\overline{f}(w)\|^2$ combined with (a):

$$\tfrac{d}{dt}V(w_t) = -\overline{f}(w_t)^{\mathsf{T}}A(w_t)[\varepsilon I + A(w_t)^{\mathsf{T}}A(w_t)]^{-1} A(w_t)^{\mathsf{T}}\overline{f}(w_t)$$

The right hand side is non-positive when $w_t \neq \theta^*$. Integrating each side gives for any $T > 0$,

$$V(w_T) = V(w_0) - \int_0^T \overline{f}(w_t)^{\mathsf{T}}A(w_t)[\varepsilon I + A(w_t)^{\mathsf{T}}A(w_t)]^{-1} A(w_t)^{\mathsf{T}}\overline{f}(w_t)\,dt \tag{43}$$

so that $V(w_T) \leq V(w_0)$ for all $T$. Under the coercive assumption, it follows that solutions to (7) are bounded. Also, letting $T \to \infty$, we obtain from (43) the bound

$$\int_0^\infty \overline{f}(w_t)^{\mathsf{T}}A(w_t)[\varepsilon I + A(w_t)^{\mathsf{T}}A(w_t)]^{-1} A(w_t)^{\mathsf{T}}\overline{f}(w_t)\,dt \leq V(w_0)$$

This combined with boundedness of $w_t$ implies that $\lim_{t\to\infty} A(w_t)^\intercal \overline{f}(w_t) = 0$.

We next prove (iii). Global asymptotic stability of (7) requires that solutions converge to $\theta^*$ from each initial condition, and also that $\theta^*$ is stable in the sense of Lyapunov [26]. Assumption (c) combined with (ii) gives the former, that $\lim_{t\to\infty} w_t = \theta^*$. A convenient sufficient condition for the latter is obtained by considering $A_\infty = \partial_\theta[\mathcal{G}(\theta)\overline{f}(\theta)] \mid_{\theta=\theta^*}$. Stability in the sense of Lypaunov holds if this matrix is Hurwitz (all eigenvalues are in the strict left half plane in $\mathbb{C}$) [26, Thm. 4.7]. Apply the definitions, we obtain $A_\infty = -[\varepsilon I + M]^{-1}M$ with $M = A(\theta^*)^\intercal A(\theta^*) > 0$ (recall that $A(\theta^*)$ is assumed to be non-singular). The matrix $A_\infty$ is negative definite, and hence Hurwitz.

$\square$

Prop. A.5 cannot be applied to the ODE (7) that motivated Zap Q-learning because $\overline{f}$ is only piecewise smooth. To obtain an extension we consider the ODE in its integral form:

$$w_t = w_0 - \int_0^t [\varepsilon I + A^\intercal(w_\tau)A(w_\tau)]^{-1} A^\intercal(w_\tau)\overline{f}(w_\tau)\, d\tau, \qquad t \geq 0 \tag{44}$$

where $\overline{f}(\theta)$, $A(\theta)$ are defined in (18, 20).

**Proposition A.6.** *Under Assumptions A1-A2, there exists a solution to* (44) *from each initial condition. The following hold for any solution:*

(i) $\overline{f}(w_t) = \overline{f}(w_0) - \int_0^t A(w_\tau)[\varepsilon I + A(w_\tau)^\intercal A(w_\tau)]^{-1} A(w_\tau)^\intercal \overline{f}(w_\tau)\, d\tau, \qquad t \geq 0$

(ii) $\|\overline{f}(w_t)\|$ *is non-increasing, and* $\lim_{t\to\infty} \overline{f}(w_t) = 0$.

(iii) *If in addition A3 holds, then the ODE* (44) *is globally asymptotically stable.*

The proof of existence is obtained by considering smooth approximations of (7).

**Lemma A.7.** *Under Assumptions A1-A2, there exists a solution to* (44) *from each initial condition.*

*Proof.* Define a $C^\infty$ probability density $\eta$ on $\mathbb{R}^d$ via

$$\eta(x) := \begin{cases} k \exp(-(1 - \|x\|^2)^{-1}) & \|x\| < 1, \\ 0 & \|x\| \geq 1, \end{cases} \tag{45}$$

where $k > 0$ is a normalization constant: $\int \eta(x)\, dx = 1$. For each $\delta > 0$, a $C^\infty$ vector field is defined via the convolution:

$$\overline{f}_\delta(x) = \frac{1}{\delta^d} \int \overline{f}(x - y)\eta(y/\delta)\, dy, \qquad x \in \mathbb{R}^d \tag{46}$$

The family of functions $\{\overline{f}_\delta : 0 < \delta \leq 1\}$ is globally uniformly Lipschitz continuous, with the same Lipschitz constant $b_L$ as of $\overline{f}$. It is also evident that $\lim_{\delta\downarrow 0} \overline{f}_\delta = \overline{f}$ pointwise. The uniform Lipschitz continuity implies that the convergence is uniform on compact sets.

Denote $A_\delta(\theta) = \partial_\theta \overline{f}_\delta(\theta)$, and consider the ODE (44) with $\overline{f}$ and $A$ replaced by their smooth approximations:

$$w_t^\delta = w_0^\delta - \int_0^t [\varepsilon I + A_\delta^\intercal(w_\tau^\delta)A_\delta(w_t^\delta)]^{-1} A_\delta^\intercal(w_t^\delta)\overline{f}_\delta(w_t^\delta)\, d\tau, \qquad w_0^\delta = w_0 \tag{47}$$

The solution exists and is unique for each $\delta \in (0, 1]$. To obtain bounds on the solution we require bounds on the matrices involved, and opt for the spectral norm:

$$\|[\varepsilon I + A_\delta^\intercal(w_t^\delta)A_\delta(w_t^\delta)]^{-1}\| = \frac{1}{\lambda_{\min}(\varepsilon I + A_\delta^\intercal(w_t^\delta)A_\delta(w_t^\delta))} \leq \frac{1}{\varepsilon}$$

$$\|A_\delta(w_\tau^\delta)\| \leq b_L$$

where $b_L$ is the Lipschitz constant for $\overline{f}_\delta$. Therefore,

$$
\begin{aligned}
\|w_t^\delta\| &\leq \|w_0^\delta\| + \int_0^t \|[\varepsilon I + A_\delta^\mathsf{T}(w_\tau^\delta)A_\delta(w_\tau^\delta)]^{-1}\| \cdot \|A_\delta(w_\tau^\delta)\| \cdot \|\overline{f}_\delta(w_\tau^\delta)\| \, d\tau \\
&\leq \|w_0^\delta\| + \frac{b_L}{\varepsilon} \int_0^t \|\overline{f}_\delta(w_\tau^\delta)\| \, d\tau \\
&\leq \|w_0^\delta\| + \frac{b_L}{\varepsilon} \int_0^t \|\overline{f}_\delta(w_0^\delta)\| + \|\overline{f}_\delta(w_\tau^\delta) - \overline{f}_\delta(w_0^\delta)\| \, d\tau \\
&\leq \|w_0^\delta\| + \frac{b_L}{\varepsilon} \left\{ T\|\overline{f}_\delta(w_0^\delta)\| + b_L \int_0^t \|w_\tau^\delta - w_0^\delta\| \, d\tau \right\} \\
&\leq \|w_0^\delta\| + \frac{b_L}{\varepsilon} \left\{ T(\|\overline{f}_\delta(w_0^\delta)\| + b_L\|w_0^\delta\|) + b_L \int_0^t \|w_\tau^\delta\| \, d\tau \right\}
\end{aligned}
$$

The set $\{\|\overline{f}_\delta(w_0^\delta)\| : 0 < \delta \leq 1\}$ is bounded by $\max_{y \in \mathcal{B}(w_0,1)} \|\overline{f}(y)\|$, where $\mathcal{B}(w_0, 1)$ denotes the closed unit ball in $\mathbb{R}^d$ centered at $w_0$. By Gronwall's inequality, there exist constants $C_1$ and $C_2$ such that

$$
\|w_t^\delta\| \leq C_1 + C_2 e^{b_L^2 T/\varepsilon}, \quad t \in [0,T], \qquad \delta \in (0,1]
$$

This combined with (47) implies that $\{w^\delta : 0 < \delta \leq 1\}$ is uniformly bounded and equicontinuous. By the Arzelà-Ascoli theorem, there exists a sequence $\delta_n \downarrow 0$ and a continuous function $w^0 : [0,T] \to \mathbb{R}^d$ such that

$$
\lim_{n \to \infty} \sup_{t \in [0,T]} \|w_t^{\delta_n} - w_t^0\| = 0
$$

So the functional equation (47) holds for $w^0$ with $\delta = 0$, and $w^0$ is thus a solution of (44). $\qquad \square$

The following result has been derived in [15, Lemma A.10]. We present it here for completeness.

**Lemma A.8.** *Let $G(\theta) := \max_{1 \leq i \leq \ell_u} G_i(\theta)$ where each $G_i : \mathbb{R}^d \to \mathbb{R}$ is twice continuously differentiable and Lipschitz continuous. Let $w : [0,T] \to \mathbb{R}^d$ be a Lipschitz continuous function, and denote $g_t := G(w_t)$. Then,*

(i) *$g : [0,T] \to \mathbb{R}$ is Lipschitz continuous.*

(ii) *At any time $t_0 \in (0,T)$ such that the derivatives of $g_t$ and $w_t$ exist,*

$$
\frac{d}{dt} g_t \Big|_{t=t_0} = \partial_\theta G_k(w_{t_0}) \cdot \frac{d}{dt} w_t \Big|_{t=t_0} \quad \text{for each } k \in \arg\max_i G_i(w_{t_0}). \tag{48}
$$

*Proof.* Denote $g_t^i = G_i(w_t)$, so that $g_t = \max_{1 \leq i \leq \ell_u} g_t^i$. Let $b_L$ denote a Lipschitz constant for each of these functions:

$$
|g_{t_1}^i - g_{t_0}^i| \leq b_L |t_1 - t_0|, \qquad t_0, t_1 \in [0,T], \quad 1 \leq i \leq \ell_u
$$

For any $t_0, t_1 \in [0,T]$,

$$
\begin{aligned}
g_{t_1} - g_{t_0} &\leq g_{t_1}^k - g_{t_0}^k, \quad \text{for each } k \in \arg\max_i g_{t_0}^i \\
&\leq b_L |t_1 - t_0|
\end{aligned}
$$

The same inequality holds for $g_{t_0} - g_{t_1}$ with $k \in \arg\max_i g_{t_1}^i$. This proves (i).

The proof of (ii) is also straightforward: The difference $g_t - g_t^k$ has a global minimum at $t_0$ if $k \in \arg\max_i g_{t_0}^i$, and consequently

$$
0 = \frac{d}{dt}[g_t - g_t^k]\big|_{t=t_0}
$$

$\qquad \square$

Given a parameter vector $\theta \in \mathbb{R}^d$, denote by $\varsigma^\theta : \mathsf{X} \times \mathsf{U} \to \mathbb{R}$ the reward function that satisfies the Bellman equation (16), with $Q^*$ replaced by $Q^\theta$: For each $x \in \mathsf{X}$ and $u \in \mathsf{U}$,

$$\varsigma^\theta(x, u) := -\gamma \sum_{x' \in \mathsf{X}} P_u(x, x') \underline{Q}^\theta(x') + Q^\theta(x, u) \tag{49}$$

**Lemma A.9.** *Suppose Assumptions A1-A2 hold and the function $w : [0, T] \to \mathbb{R}^d$ is Lipschitz continuous. Then, $\varsigma^{w_t}(x, u)$ is Lipschitz continuous in $t$ for each $x, u$. Moreover, at any point $t_0$ of differentiability,*

$$\frac{d}{dt} \varsigma^{w_t}(x, u) \Big|_{t=t_0} = \Big[ -\gamma \sum_{x' \in \mathsf{X}} P_u(x, x') \partial_\theta Q^{w_{t_0}}(x', \phi^{w_{t_0}}(x')) + \partial_\theta Q^{w_{t_0}}(x, u) \Big] \frac{d}{dt} w_t \Big|_{t=t_0} \tag{50}$$

*where $\phi^{w_{t_0}}$ is defined in (17).*

*Proof.* From the definition (49), it is sufficient to establish the derivative formula

$$\frac{d}{dt} \underline{Q}^{w_t}(x') \Big|_{t=t_0} = \partial_\theta Q^{w_{t_0}}(x', \phi^{(k)}(x')) \cdot \frac{d}{dt} w_t \Big|_{t=t_0}$$

where $\phi^{(k)}$ is *any* policy that is $Q^{w_{t_0}}$-greedy. This is immediate from Lemma A.8. $\square$

Stability of is obtained from the following standard Lyapunov condition:

**Lemma A.10.** *Suppose that $\{w_t : t \in \mathbb{R}\}$ is a Lipschitz continuous function taking values in $\mathbb{R}^d$, $V : \mathbb{R}^d \to \mathbb{R}_+$ is continuous and coercive, and $U : \mathbb{R}^d \to \mathbb{R}_+$. Assume moreover the following properties:*

(i) $\inf\{U(\theta) : V(\theta) \geq \delta\} > 0$ *for each $\delta > 0$.*

(ii) $V(w_t) \leq V(w_0) - \displaystyle\int_0^t U(w_\tau) \, d\tau$, $\qquad t \geq 0$.

*Then, there exists a function $B : \mathbb{R}_+ \to \mathbb{R}_+$ such that $V(w_t) \leq \eta$ for all $t \geq V(w_0) B(\eta)$. In particular, $V(w_t) \to 0$ as $t \to \infty$.*

*Proof.* For any scalar $\eta$ satisfying $0 < \eta < V(w_0)$, let $H^\eta := \{\theta : \eta \leq \theta \leq V(w_0)\}$ and

$$\varepsilon_\eta = \inf_{\theta \in H^\eta} U(\theta)$$

Under assumption (i) of the lemma we have $\varepsilon_\eta > 0$. Let $T^\eta = \inf\{t : V(w_t) \leq \eta\}$, so that $w_t \in H^\eta$ for $0 \leq t \leq T^\eta$. By assumption (ii) we have

$$0 \leq V(w_t) \leq V(w_0) - \varepsilon_\eta t, \qquad 0 < t \leq T^\eta.$$

Therefore, $T^\eta < V(w_0)/\varepsilon_\eta$. Because $V(w_t)$ is non-increasing in $t$, we have $V(w_t) \leq \eta$ for all $t \geq V(w_0) B(\varepsilon)$, with $B(\varepsilon) = \varepsilon_\eta^{-1}$.

Since $\eta$ is arbitrary, it follows that $\lim_{t \to \infty} V(w_t) = 0$. $\square$

*Proof of Prop. A.6.* Suppose $w : [0, T] \to \mathbb{R}^d$ is a solution of (44). At point $t$ of differentiability, the derivative of $\overline{f}(w_t)$ is given by

$$\begin{aligned}
\frac{d}{dt} \overline{f}(w_t) &= \frac{d}{dt} \mathsf{E}\Big[ \zeta_n \varsigma^{w_t}(X_n, U_n) \Big] \\
&= \mathsf{E}[\zeta_n \frac{d}{dt} \varsigma^{w_t}(X_n, U_n)] + \mathsf{E}[\mathcal{D}(w_t, \Phi_{n+1}) \frac{d}{dt} \zeta_n]
\end{aligned} \tag{51}$$

For each $x \in \mathsf{X}$ and $u \in \mathsf{U}$, $\varsigma^{w_t}(x, u)$ is a Lipschitz continuous function of $t$, whose derivative is given in Lemma A.9. Assertion (i) follows:

$$\begin{aligned}
\frac{d}{dt} \overline{f}(w_t) &= \mathsf{E}\Big[ \zeta_n [\gamma \partial_\theta Q^{w_t}(X_{n+1}, \phi^{w_t}(X_{n+1})) - \partial_\theta Q^{w_t}(X_n, U_n)] + \mathcal{D}(w_t, \Phi_{n+1}) \partial_\theta \zeta_n \Big] \frac{d}{dt} w_t \\
&= -A(w_t)[\varepsilon I + A(w_t)^\intercal A(w_t)]^{-1} A(w_t)^\intercal \overline{f}(w_t)
\end{aligned}$$

A candidate Lyapunov function is defined as $V(w_t) := \frac{1}{2}\|\overline{f}(w_t)\|^2$. At a point $t$ where $\overline{f}(w_t)$ is differentiable,

$$\tfrac{d}{dt}V(w_t) = -\overline{f}(w_t)^\mathsf{T} A(w_t)[\varepsilon I + A(w_t)^\mathsf{T} A(w_t)]^{-1}A(w_t)^\mathsf{T}\overline{f}(w_t) \tag{52}$$

$$-[\varepsilon I + A(\theta)^\mathsf{T} A(\theta)]^{-1} \le -b_V I$$

The integral representation of (52) then gives, for any $t \in [0, T]$,

$$V(w_t) = V(w_0) - \int_0^t \overline{f}(w_\tau)^\mathsf{T} A(w_\tau)[\varepsilon I + A(w_\tau)^\mathsf{T} A(w_\tau)]^{-1}A(w_\tau)^\mathsf{T}\overline{f}(w_\tau)\, d\tau$$

$$\le V(w_0) - b_V \int_0^t \|A(w_\tau)^\mathsf{T}\overline{f}(w_\tau)\|^2\, d\tau \tag{53}$$

Under (A2) the assumptions of Lemma A.10 hold with $U(\theta) = b_V\|A(\theta)^\mathsf{T}\overline{f}(\theta)\|^2$, so that $\lim_{t\to\infty} V(w_t) = \lim_{t\to\infty} \overline{f}(w_t) = 0$.

If in addition (A3) holds, we conclude that $\lim_{t\to\infty} w_t = \theta^*$. $\qquad\square$

## A.5 Proof of Thm. 2.1

The remainder of the Appendix is dedicated to the proof of Thm. 2.1. We use $n_0 = 0$ in the definition of the step-size sequences (22); this shortens many of the expressions that follow, and the extension to general $n_0 \ge 1$ is obvious. Given the typical choice of $\zeta_n$ in (2), it is assumed throughout that $\zeta_n := \zeta(\theta_n, X_n, U_n)$ for some function $\zeta : \mathbb{R}^d \times \mathsf{X} \times \mathsf{U} \to \mathbb{R}^d$. We proceed under the additional assumption that the vector-valued function $\zeta$ has non-negative entries:

$$[\zeta(\theta, x, u)]_i \ge 0 \text{ for each } i, \theta, x, u. \tag{54}$$

The proofs are extended to the general case in Section A.5.4.

### A.5.1 Generalities

This subsection contains the building blocks of the proof, summarized in two propositions, and the proof of Thm. 2.1 based on these key results. The proofs of the propositions are postponed to subsequent subsections.

The *slow time scale* used for an ODE approximation of $\{\theta_n\}$ is defined by

$$t_n = \sum_{i=1}^n \alpha_i = \sum_{i=1}^n \frac{1}{i}, \qquad n \ge 1, \qquad t_0 = 0 \tag{55}$$

and its approximate inverse

$$[t] := \max\{j : t_j \le t\} \tag{56}$$

Define the continuous time process $\{\bar{w}_t : t \ge 0\}$ with $\bar{w}_{t_n} = \theta_n$, and extended to $\mathbb{R}_+$ via linear interpolation. Define the associated continuous time process $\{\bar{c}_t := \overline{f}(\bar{w}_t) : t \ge 0\}$. We also define the piecewise constant processes $\{\bar{\mathcal{A}}_t, \bar{\mathcal{G}}_t : t \ge 0\}$ with $\bar{\mathcal{A}}_t = \widehat{A}_{n+1}, \bar{\mathcal{G}}_t = G_{n+1}$ for $t \in [t_n, t_{n+1})$. Both $b_\theta := \sup_n \|\theta_n\| = \sup_t \|\bar{w}_t\|$ and $b_c := \sup_t \|\bar{c}_t\|$ are finite *a.s.* by assumption.

Denote by $\mathcal{O}(1) = e(T_0, t)$ a function of two variables, satisfying for each $T > 0$,

$$\lim_{T_0\to\infty} \sup_{0\le t\le T} \|e(T_0, t)\| = 0$$

**Proposition A.11.** *Under Assumptions (A1)-(A2) and (54), $\{\bar{w}_t\}$ and $\{\bar{c}_t\}$ are Lipschitz continuous with respect to $t$, and the following approximations hold:*

(i) $\displaystyle \lim_{T_0\to\infty} \int_{T_0}^{T_0+T} \|\bar{\mathcal{A}}_t \tfrac{d}{dt}\bar{w}_t - \tfrac{d}{dt}\bar{c}_t\|_\infty\, dt = 0.$

(ii) $\displaystyle \bar{c}_{T_0+t} = \bar{c}_{T_0} + \int_{T_0}^{T_0+t} \bar{\mathcal{A}}_\tau \bar{\mathcal{G}}_\tau \bar{c}_\tau\, d\tau + \mathcal{O}(1), \qquad T_0 \to \infty$

(iii) $\|\bar{c}_{T_0+t}\|^2 = \|\bar{c}_{T_0}\|^2 + 2\int_{T_0}^{T_0+t} \bar{c}_\tau^\mathsf{T}\bar{\mathcal{A}}_\tau\bar{\mathcal{G}}_\tau\bar{c}_\tau \, d\tau + o(1)\,, \qquad T_0 \to \infty. \qquad\qquad \square$

For a fixed but arbitrary time-horizon $T > 0$, define a family of functions $\{\overline{\Gamma}^{T_0} : T_0 \geq 0\}$, where $\overline{\Gamma}^{T_0} : [0,T] \to \mathbb{R}^m$ for each $T_0 \geq 0$ and an integer $m$. It consists of four components: for $t \in [0,T]$,

$$\overline{\Gamma}_1^{T_0}(t) = \bar{w}_{T_0+t}\,, \qquad \overline{\Gamma}_2^{T_0}(t) = \bar{c}_{T_0+t}\,, \qquad \overline{\Gamma}_3^{T_0}(t) = \bar{\mathcal{A}}_{T_0+t}\,, \qquad \overline{\Gamma}_4^{T_0}(t) = -\bar{\mathcal{A}}_{T_0+t}\bar{\mathcal{G}}_{T_0+t}$$

$\{\overline{\Gamma}_1^{T_0} : T_0 \geq 0\}$ and $\{\overline{\Gamma}_2^{T_0} : T_0 \geq 0\}$ are uniformly Lipschitz continuous and bounded. More specifically, each of $\overline{\Gamma}_1^{T_0}$ and $\overline{\Gamma}_2^{T_0}$ is a function of two variables: $\overline{\Gamma}_1^{T_0}(\omega, t), \overline{\Gamma}_2^{T_0}(\omega, t)$ with $\omega \in \Omega$ and $t \in \mathbb{R}_+$. The property that $\overline{\Gamma}_1^{T_0}$ and $\overline{\Gamma}_2^{T_0}$ are Lipschitz continuous and bounded holds with probability one. Denote their sub-sequential limits by

$$\Gamma_1(t) = w_t\,, \qquad \Gamma_2(t) = c_t$$

where the convergence is uniform over $[0,T]$.

Limits of the remaining components of $\Gamma$ are defined with respect to the weak topology in $L_2([0,T]; \mathbb{R}^{d\times d})$. Because $\{\overline{\Gamma}_3^{T_0} : T_0 \geq 0\}$ and $\{\overline{\Gamma}_4^{T_0} : T_0 \geq 0\}$ are uniformly bounded, they are weakly relatively sequentially compact in $L_2([0,T]; \mathbb{R}^{d\times d})$ [17, Theorem 1.1.2]. Their weak sub-sequential limits $\Gamma_3$ and $\Gamma_4$ are denoted by $\{\mathcal{A}_t, \mathcal{H}_t : 0 \leq t \leq T\}$. That is, there exists $T_k \to \infty$ such that

$$\overline{\Gamma}_3^{T_k} \to \mathcal{A} \text{ weakly in } L_2([0,T]; \mathbb{R}^{d\times d})\,, \qquad \overline{\Gamma}_4^{T_k} \to \mathcal{H} \text{ weakly in } L_2([0,T]; \mathbb{R}^{d\times d})\,, \qquad k \to \infty$$

Based on Prop. A.11, and a separate analysis of the *fast time scale* recursion for $\{\widehat{A}_n\}$ we obtain the following properties for any sub-sequential limit $\Gamma$ of $\{\overline{\Gamma}^{T_0} : T_0 \geq 0\}$:

**Proposition A.12.** *Under Assumptions (A1)-(A2) and (54), for each $t \in [0,T]$,*

(i) $c_t := \Gamma_2(t) = \overline{f}(w_t)$.

(ii) $\mathcal{A}_t := \Gamma_3(t) \in \mathcal{A}(w_t)$.

(iii) $\mathcal{H}_t := \Gamma_4(t) \in \mathbb{R}^{d\times d}$ *is positive semi-definite.*

(iv) *There exists a constant $b_V > 0$ such that, for a.e. $t \in [0,T]$,*

$$\frac{d}{dt}c_t = -\mathcal{H}_t c_t \tag{57a}$$

$$\frac{d}{dt}V(w_t) \leq -U(w_t) \tag{57b}$$

*with $V(w_t) = \frac{1}{2}\|\overline{f}(w_t)\|^2$ and $U(w_t) = b_V\|\mathcal{A}_t^\mathsf{T}c_t\|^2$.* $\qquad\qquad \square$

An alert reader will notice that we have *not* obtained the desired ODE limit, since (57a) may differ from the ODE solution given in Lemma A.6 (i). In particular, we do not know if $\mathcal{A}_t$ coincides with $A(w_t)$ (where $A(\theta)$ is defined in (20) using a particular $Q^\theta$-greedy policy), and we do not know if $\mathcal{H}_t$ coincides with

$$A(w_t)[\varepsilon I + A(w_t)^\mathsf{T}A(w_t)]^{-1}A(w_t)^\mathsf{T}$$

We preserve the essential drift condition (57b), which leads to a simple proof of the main result:

*Proof of Thm. 2.1.* Prop. A.12 (i) and (ii) justify the assertion that $U(w_t) := b_V\|\mathcal{A}_t^\mathsf{T}c_t\|^2$ is in fact a function of $w_t$. Under (A2) we see that Assumption (i) of Lemma A.10 holds, and (57b) implies Assumption (ii) of the lemma.

For given $\eta > 0$, we may choose $T \geq V(w_0)B(\eta)$, so that $V(w_T) \leq \eta$ for any sub-sequential limit. It then follows that $\limsup_{n\to\infty} V(\theta_n) \leq \eta$. Since $\eta > 0$ is arbitrary, it follows that $V(w_T) \equiv 0$; that is, $\lim_{t\to\infty} \overline{f}(\theta_n) = 0$ as claimed. $\qquad\qquad \square$

### A.5.2 Analysis of $\{\widehat{A}_n\}$ over the fast time scale

The goal in this subsection is to show that $\widehat{A}_n$ is close to the set $\mathcal{A}(\theta_n)$ with $n$ sufficiently large. An explicit representation of $\mathcal{A}(\theta)$ is given in the following: denote, for any $\theta \in \mathbb{R}^d$ and any (possibly randomized) policy $\phi$, the random $d \times d$ matrix:

$$A_{n+1}(\theta, \phi) = \big[\gamma \partial_\theta Q^\theta(X_{n+1}, \phi(X_{n+1})) - \partial_\theta Q^\theta(X_n, U_n)\big]\zeta(\theta, X_n, U_n)$$

If $\phi$ is $Q^\theta$-greedy, meaning

$$Q^\theta(x, \phi(x)) = \underline{Q}^\theta(x), \qquad x \in \mathsf{X},$$

then a generalized subgradient of the function $f$ in (19b) is given by

$$A_{n+1}(\theta, \phi) + \mathcal{D}(\theta, \Phi_{n+1})\partial_\theta \zeta(\theta, X_n, U_n)$$

**Lemma A.13.** *If (A1)-(A2) hold, and if $\zeta$ is non-negative, then the set $\mathcal{A}(\theta)$ defined in (26) admits the representation,*

$$\mathcal{A}(\theta) = \big\{\mathsf{E}_\varpi[A_{n+1}(\theta, \tilde{\phi}_{n+1}^\theta) + \mathcal{D}(\theta, \Phi_{n+1})\partial_\theta \zeta(\theta, X_n, U_n)] \ : \ \tilde{\phi}_{n+1}^\theta \text{ is } Q^\theta\text{-greedy}\big\}$$

*where $\tilde{\phi}_{n+1}^\theta$ ranges over all $Q^\theta$-greedy randomized policies.* $\qquad\qquad\square$

A key implication of the non-negativity assumption (54) is the following:

**Lemma A.14.** *Under Assumptions (A1)-(A2) and (54), there exists $b_T < \infty$ such that, for all $n \geq 1$ and all vectors $v \in \mathbb{R}^d$, $\|v\| \leq 1$,*

$$f(\theta_n + v, \Phi_{n+1}) \geq f(\theta_n, \Phi_{n+1}) + A_{n+1}v - b_T\|v\|^2\mathbf{1} \tag{58}$$

*where the inequality is component-wise, $A_{n+1}$ is defined in (21a), and $\mathbf{1}$ denotes the vector of all ones. In particular, when $Q^\theta = \psi^\intercal\theta$, we have $b_T = 0$:*

$$f(\theta_n + v, \Phi_{n+1}) \geq f(\theta_n, \Phi_{n+1}) + A_{n+1}v \tag{59}$$

*Proof.* The proof is based on the Taylor series expansion. With $z := (x', x, u', u)$, define $g : \mathbb{R}^d \times \mathsf{Z} \times \mathsf{U} \to \mathbb{R}$ by

$$g(\theta, z, u^\circ) := r(x, u) + \gamma Q^\theta(x', u^\circ) - Q^\theta(x, u) \tag{60}$$

By (A2), $g$ admits the Taylor series expansion at each $\|\theta\| \leq b_\theta$:

$$g(\theta + v, z, u^\circ) = g(\theta, z, u^\circ) + \partial_\theta g(\theta, z, u^\circ)v + O(\|v\|^2)$$

Recall that $\mathcal{D}(\theta, z) := g(\theta, z, \phi^\theta(x')) = \max_{u^\circ} g(\theta, z, u^\circ)$ and the state-input space is finite,

$$\begin{aligned}
\mathcal{D}(\theta_n + v, \Phi_{n+1}) &= \max_{u^\circ} g(\theta_n + v, \Phi_{n+1}, u^\circ) \\
&= \max_{u^\circ} g(\theta_n, \Phi_{n+1}, u^\circ) + \partial_\theta g(\theta_n, \Phi_{n+1}, u^\circ)v + O(\|v\|^2) \\
&\geq \mathcal{D}(\theta_n, \Phi_{n+1}) + \partial_\theta g(\theta_n, \Phi_{n+1}, \phi^{\theta_n}(X_{n+1}))v + O(\|v\|^2)
\end{aligned} \tag{61}$$

Denote $\zeta_n(\theta) := \zeta(\theta, X_n, U_n)$. Another Taylor series expansion of $\zeta$ at $\theta_n$ gives

$$\zeta_n(\theta_n + v) = \zeta_n(\theta_n) + \partial_\theta \zeta_n(\theta_n)v + O(\|v\|^2) \tag{62}$$

We next recall that $f(\theta_n, \Phi_{n+1}) = \zeta_n(\theta_n)\mathcal{D}(\theta_n, \Phi_{n+1})$,

$$\begin{aligned}
f(\theta_n + v, \Phi_{n+1}) - f(\theta_n, \Phi_{n+1}) =&\zeta_n(\theta_n)\big\{\mathcal{D}(\theta_n + v, \Phi_{n+1}) - \mathcal{D}(\theta_n, \Phi_{n+1})\big\} \\
&+ \big\{\zeta_n(\theta_n + v) - \zeta_n(\theta_n)\big\}\mathcal{D}(\theta_n, \Phi_{n+1}) \\
&+ \big\{\zeta_n(\theta_n + v) - \zeta_n(\theta_n)\big\}\big\{\mathcal{D}(\theta_n + v, \Phi_{n+1}) - \mathcal{D}(\theta_n, \Phi_{n+1})\big\}
\end{aligned}$$

By (61) and the non-negativity assumption (54),

$$\zeta_n(\theta_n)\big\{\mathcal{D}(\theta_n + v, \Phi_{n+1}) - \mathcal{D}(\theta_n, \Phi_{n+1})\big\} \geq \zeta_n(\theta_n)\partial_\theta g(\theta_n, \Phi_{n+1}, \phi^{\theta_n}(X_{n+1}))v + O(\|v\|^2)$$

Similarly, from (62),

$$\{\zeta_n(\theta_n + v) - \zeta_n(\theta_n)\}\mathcal{D}(\theta_n, \Phi_{n+1}) = \{\partial_\theta\zeta_n(\theta_n)v + O(\|v\|^2)\}\mathcal{D}(\theta_n, \Phi_{n+1})$$
$$\geq \mathcal{D}(\theta_n, \Phi_{n+1})\partial_\theta\zeta_n(\theta_n)v + O(\|v\|^2)$$

By (A2) once more, both $\zeta$ and $\mathcal{D}$ are Lipschitz continuous in $\theta$,

$$\left\|\zeta_n(\theta_n + v) - \zeta_n(\theta_n)\right\|\left|\mathcal{D}(\theta_n + v, \Phi_{n+1}) - \mathcal{D}(\theta_n, \Phi_{n+1})\right| = O(\|v\|^2)$$

Consequently,

$$f(\theta_n + v, \Phi_{n+1}) - f(\theta_n, \Phi_{n+1})$$
$$\geq \left\{\zeta_n(\theta_n)\partial_\theta g(\theta_n, \Phi_{n+1}, \phi^{\theta_n}(X_{n+1})) + \mathcal{D}(\theta_n, \Phi_{n+1})\partial_\theta\zeta_n(\theta_n)\right\}v + O(\|v\|^2)$$

The proof is completed by realizing that $A_{n+1}$ defined in (21a) can be expressed

$$A_{n+1} = \zeta_n(\theta_n)\partial_\theta g(\theta_n, \Phi_{n+1}, \phi^{\theta_n}(X_{n+1})) + \mathcal{D}(\theta_n, \Phi_{n+1})\partial_\theta\zeta_n(\theta_n)$$

□

Define the *fast time scale*, over which the matrix gain sequence $\{\widehat{A}_n\}$ is updated,

$$t_n = \sum_{i=1}^{n}\beta_i = \sum_{i=1}^{n}\frac{1}{i^\rho}, \qquad n \geq 1, \qquad t_0 = 0, \qquad \rho \in (0.5, 1) \tag{63}$$

Define the time process $\{\bar{\mathcal{A}}_t : t \geq 0\}$ with $\bar{\mathcal{A}}_{t_n} = \widehat{A}_n$ for those values $t_n$, with the definition extended to $\mathbb{R}_+$ via linear interpolation. Note that this definition of $\{\bar{\mathcal{A}}_t : t \geq 0\}$ is used only in this subsection to analyze $\{\widehat{A}_n\}$. For each $n \geq 1$, define the associated time block: $[t_{m(n)}, t_n)$ where $m(n) = \min\{j : t_j + \ln(n) \geq t_n\}$. Some properties of this fast time scale setting are collected in the following:

**Lemma A.15.** *The follow hold:*

(i) $\ln(n) - 1 < t_n - t_{m(n)} \leq \ln(n)$.

(ii) *There exists $N_s \geq 1$ such that for $n \geq N_s$, $m(n) + 1 \geq \rho^{1/(1-\rho)}(n+1)$.*

(iii) $\lim_{n\to\infty}\max_{m(n)\leq k\leq n}\|\theta_k - \theta_n\| = 0$.

*Proof.* (i) follows directly from the definition.

By (63),

$$t_n - t_{m(n)} = \sum_{i=m(n)+1}^{n}\frac{1}{i^\rho} \geq \int_{m(n)+1}^{n+1}\frac{1}{\tau^\rho}d\tau$$
$$= (1-\rho)^{-1}[(n+1)^{1-\rho} - (m(n)+1)^{1-\rho}] \tag{64}$$

Since $\ln(n) \geq t_n - t_{m(n)}$, we have

$$(1-\rho)\ln(n) \geq (n+1)^{1-\rho} - (m(n)+1)^{1-\rho}$$

There exits $N_s \geq 1$ such that $(n+1)^{1-\rho} \geq \ln(n)$ for $n \geq N_s$. Hence,

$$(1-\rho)(n+1)^{1-\rho} \geq (n+1)^{1-\rho} - (m(n)+1)^{1-\rho}, \qquad n \geq N_s$$

which proves (ii).

By (64),

$$(1-\rho)^{-1}[(n+1)^{1-\rho} - (k+1)^{1-\rho}] \leq (1-\rho)^{-1}[(n+1)^{1-\rho} - (m(n)+1)^{1-\rho}]$$
$$\leq \ln(n)$$

Multiplying each side of above inequality by $(1-\rho)(k+1)^{\rho-1}$ gives

$$\left(\frac{n+1}{k+1}\right)^{1-\rho} - 1 \leq (1-\rho)(k+1)^{\rho-1}\ln(n) \leq (1-\rho)(m(n)+1)^{\rho-1}\ln(n)$$

By the inequality $\ln(1 + x) \le x$ for $x > -1$,

$$(1 - \rho)\ln\Big(\frac{n+1}{k+1}\Big) \le \ln\big(1 + (1-\rho)(m(n)+1)^{\rho-1}\ln(n)\big) \le (1-\rho)(m(n)+1)^{\rho-1}\ln(n)$$

Given $m(n) + 1 \ge \rho^{1/(1-\rho)}(n+1)$ in (ii),

$$\ln\Big(\frac{n+1}{k+1}\Big) \le \rho^{-1}\ln(n)(n+1)^{\rho-1} \tag{65}$$

The parameter vector $\theta_n$ updated by (21d) can be expressed

$$\theta_n = \theta_k + \sum_{i=k+1}^{n} \alpha_i G_i f(\theta_{i-1}, \Phi_i), \qquad m(n) \le k < n$$

We can find a constant $b_f < \infty$ such that $\sup_n \|G_{n+1} f(\theta_n, \Phi_{n+1})\| \le b_f$ for almost every $\omega \in \Omega$. With $\alpha_i \equiv 1/i$,

$$\|\theta_n - \theta_k\| \le b_f \sum_{i=k+1}^{n} \alpha_i \le b_f \int_k^n \frac{1}{\tau}\,d\tau \le b_f \ln\Big(\frac{n}{k}\Big)$$

By (65), for $n \ge N_s$.

$$
\begin{aligned}
\|\theta_n - \theta_k\| &\le b_f\Big|\ln\Big(\frac{n}{k}\Big) - \ln\Big(\frac{n+1}{k+1}\Big)\Big| + b_f\rho^{-1}\ln(n)(n+1)^{\rho-1} \\
&\le b_f\Big|\ln(1 - \frac{1}{n+1}) + \ln(1 + \frac{1}{k})\Big| + b_f\rho^{-1}\ln(n)(n+1)^{\rho-1} \\
&\le b_f\frac{1}{k} + b_f\rho^{-1}\ln(n)(n+1)^{\rho-1} \\
&\le b_f\frac{1}{\rho^{1/(1-\rho)}(n+1) - 1} + b_f\rho^{-1}\ln(n)(n+1)^{\rho-1}
\end{aligned}
\tag{66}
$$

where the last inequality holds given $k \ge m(n) \ge \rho^{1/(1-\rho)}(n+1) - 1$. Therefore, $\max_{m(n)\le k\le n} \|\theta_k - \theta_n\| \to 0$ as $n \to \infty$. □

**Proposition A.16.** *Under Assumptions (A1)-(A2) and (54), the following hold for all $v \in \mathbb{R}^d$, $\|v\| \le 1$, and all $k \in \mathbb{Z}$ between $m(n)$ and $n$:*

(i)

$$\sum_{i=k+1}^{n} \beta_i[f(\theta_{i-1} + v, \Phi_i) - f(\theta_{i-1}, \Phi_i) + b_T\|v\|^2 \mathbf{1}] \ge \widehat{A}_n v - \widehat{A}_k v + \sum_{i=k+1}^{n} \beta_i \widehat{A}_{i-1} v \tag{67}$$

(ii) *For any $t \in [t_{m(n)}, t_n)$,*

$$\bar{\mathcal{A}}_{t_n} v - \bar{\mathcal{A}}_t v + \int_t^{t_n} \bar{\mathcal{A}}_\tau v\,d\tau \le (t_n - t)[\overline{f}(\theta_n + v) - \overline{f}(\theta_n) + b_T\|v\|^2\mathbf{1}] + o(1), \qquad n \to \infty \tag{68}$$

*where $o(1) \to 0$ as $n \to \infty$, uniformly in $v$.*

*Proof.* By (58), for each $n \ge 1$,

$$f(\theta_n + v, \Phi_{n+1}) \ge f(\theta_n, \Phi_{n+1}) + A_{n+1} v - b_T\|v\|^2\mathbf{1}, \qquad v \in \mathbb{R}^d$$

Consequently,

$$\sum_{i=k+1}^{n} \beta_i[f(\theta_{i-1} + v, \Phi_i) - f(\theta_{i-1}, \Phi_i) + b_T\|v\|^2\mathbf{1}] \ge \sum_{i=k+1}^{n} \beta_i A_i v, \qquad m(n) \le k \le n$$

The gain matrix $\widehat{A}_n$ updated by (21b) can be expressed

$$\widehat{A}_n = \widehat{A}_k + \sum_{i=k+1}^{n} \beta_i A_i - \sum_{i=k+1}^{n} \beta_i \widehat{A}_{i-1}, \qquad m(n) \le k \le n$$

Therefore, $\sum_{i=k+1}^{n} \beta_i A_i v = \widehat{A}_n v - \widehat{A}_k v + \sum_{i=k+1}^{n} \beta_i \widehat{A}_{i-1} v$. This proves (i).

Now consider the sum $\sum_{i=k+1}^{n} \beta_i f(\theta_{i-1} + v, \Phi_i)$ with $m(n) \le k \le n$. We first rewrite it in the suggestive form

$$\sum_{i=k+1}^{n} \beta_i f(\theta_{i-1} + v, \Phi_i) = \sum_{i=k+1}^{n} \beta_i [f(\theta_{i-1} + v, \Phi_i) - f(\theta_n + v, \Phi_i)] + \sum_{i=k+1}^{n} \beta_i f(\theta_n + v, \Phi_i)$$

By the Lipschitz continuity of $f$ in $\theta$ and Lemma A.15 (iii), the first sum on the right hand side goes to 0 uniformly in $k$ as $n \to \infty$. The second sum can be expressed

$$\sum_{i=k+1}^{n} \beta_i f(\theta_n + v, \Phi_i) = (t_n - t_k) \overline{f}(\theta_n + v) + \sum_{i=k+1}^{n} \beta_i [f(\theta_n + v, \Phi_i) - \overline{f}(\theta_n + v)]$$

Each term in the sum on the right side has zero-mean under the stationary pmf of $(\boldsymbol{X}, \boldsymbol{U})$. It goes to zero $a.s.$ for $m(n) \le k \le n$ as $n \to \infty$ [5, Part II, Section 1.4.6, Proposition 7]. We then obtain

$$\max_{m(n) \le k \le n} \left\| (t_n - t_k) \overline{f}(\theta_n + v) - \sum_{i=k+1}^{n} \beta_i f(\theta_{i-1} + v, \Phi_i) \right\| = o(1), \qquad n \to \infty \qquad (69)$$

Since the process $\{\bar{\mathcal{A}}_t : t \ge 0\}$ is linearly interpolated between discrete values,

$$\int_{t_k}^{t_n} \bar{\mathcal{A}}_\tau v \, d\tau = \tfrac{1}{2} \sum_{i=k+1}^{n} \beta_i [\widehat{A}_i + \widehat{A}_{i-1}] v = \sum_{i=k+1}^{n} \beta_i \widehat{A}_{i-1} v + \tfrac{1}{2} \sum_{i=k+1}^{n} \beta_i [\widehat{A}_i - \widehat{A}_{i-1}] v$$

where the second sum on the right hand side can be rewritten as

$$\sum_{i=k+1}^{n} \beta_i [\widehat{A}_i - \widehat{A}_{i-1}] v = -\beta_{k+1} \widehat{A}_k v + \beta_n \widehat{A}_{n+1} v + \sum_{i=k+1}^{n-1} [\beta_i - \beta_{i+1}] \widehat{A}_i v$$

which goes to zero as $n \to \infty$ given $\sup_n \|\widehat{A}_n\| < \infty$ and $\beta_i - \beta_{i+1} \approx \rho i^{-1} \beta_i$. Therefore,

$$\max_{m(n) \le k \le n} \left\| \sum_{i=k+1}^{n} \beta_i \widehat{A}_{i-1} v - \int_{t_k}^{t_n} \bar{\mathcal{A}}_\tau v \, d\tau \right\| = o(1), \qquad n \to \infty \qquad (70)$$

Combining (i) with (69) and (70) gives, for $t \in \{t_k : m(n) \le k \le n\}$,

$$\bar{\mathcal{A}}_{t_n} v - \bar{\mathcal{A}}_t v + \int_t^{t_n} \bar{\mathcal{A}}_\tau v \, d\tau \le (t_n - t)[\overline{f}(\theta_n + v) - \overline{f}(\theta_n) + b_T \|v\|^2 \mathbf{1}] + o(1) \qquad (71)$$

For any $t \in [t_{m(n)}, t_n)$, denote $k = \max\{j : t_j \le t\}$. Letting $\delta = (t - t_k)/(t_{k+1} - t_k)$, we have

$$\bar{\mathcal{A}}_t v = (1 - \delta) \bar{\mathcal{A}}_{t_k} v + \delta \bar{\mathcal{A}}_{t_{k+1}} v$$

Then,

$$(1 - \delta) \left\{ \bar{\mathcal{A}}_{t_n} v - \bar{\mathcal{A}}_{t_k} v + \int_{t_k}^{t_n} \bar{\mathcal{A}}_\tau v \, d\tau \right\} \le (1 - \delta)(t_n - t_k)[\overline{f}(\theta_n + v) - \overline{f}(\theta_n) + b_T \|v\|^2 \mathbf{1}] + o(1)$$

$$\delta \left\{ \bar{\mathcal{A}}_{t_n} v - \bar{\mathcal{A}}_{t_{k+1}} v + \int_{t_{k+1}}^{t_n} \bar{\mathcal{A}}_\tau v \, d\tau \right\} \le \delta(t_n - t_{k+1})[\overline{f}(\theta_n + v) - \overline{f}(\theta_n) + b_T \|v\|^2 \mathbf{1}] + o(1)$$

Combining above two inequalities gives

$$\bar{\mathcal{A}}_{t_n} v - \bar{\mathcal{A}}_t v + \int_t^{t_n} \bar{\mathcal{A}}_\tau v \, d\tau \le (t_n - t)[\overline{f}(\theta_n + v) - \overline{f}(\theta_n) + b_T \|v\|^2 \mathbf{1}] + o(1)$$

$\square$

Recall the constant $b_T > 0$ introduced in Lemma A.14. For fixed matrix $\widehat{A} \in \mathbb{R}^{d \times d}$ and vector $\theta \in \mathbb{R}^d$, define the function $\text{dist}_{\mathcal{N}} : \mathbb{R}^{d \times d} \times \mathbb{R}^d \to \mathbb{R}$ by

$$\text{dist}_{\mathcal{N}}(\widehat{A}, \theta) = \sup_{\|v\| \le 1} \left\{ \max_i \left[ \widehat{A}v - (\overline{f}(\theta + v) - \overline{f}(\theta)) \right]_i - b_T \|v\|^2 \right\} \tag{72}$$

This measures how well $\widehat{A}v$ approximates the directional derivative $f'(\theta; v)$ for $v$ in the unit ball. It is non-negative since $v = 0$ is feasible in the supremum in (72). It is also continuous in both arguments:

**Proposition A.17.** *Under Assumptions (A1)-(A2) and* (54)*, the function $\text{dist}_{\mathcal{N}}$ defined in* (72) *satisfies:*

(i) *For fixed $\widehat{A}$ and $\theta$, the supremum in* (72) *is achieved.*

(ii) *$\text{dist}_{\mathcal{N}}(\widehat{A}, \theta)$ is non-negative and Lipschitz continuous in both $\widehat{A}$ and $\theta$.*

(iii) *If $\text{dist}_{\mathcal{N}}(\widehat{A}, \theta) = 0$, then the following hold: $\widehat{A} \in \mathcal{A}(\theta)$, and*

*If $\overline{f}'(\theta; v) = -\overline{f}'(\theta; -v)$ for some $\|v\| \le 1$, then $\widehat{A}v = \overline{f}'(\theta; v)$.*

*Proof.* With fixed $\widehat{A}$ and $\theta$, $\max_i [\widehat{A}v - (\overline{f}(\theta + v) - \overline{f}(\theta))]_i$ is Lipschitz continuous with respect to $v$ by Lemma A.8 (i). Since the set $\{v : \|v\| \le 1\}$ is compact, the supremum is achieved.

For (ii), consider $\widehat{A} \ne \widehat{A}'$, while $\theta$ is fixed. Let $v^*, i^*$ maximize $[\widehat{A}v - (\overline{f}(\theta + v) - \overline{f}(\theta))]_i - b_T \|v\|^2$. We have

$$\text{dist}_{\mathcal{N}}(\widehat{A}, \theta) - \text{dist}_{\mathcal{N}}(\widehat{A}', \theta) \le [\widehat{A}v^* - (\overline{f}(\theta + v^*) - \overline{f}(\theta))]_{i^*} - [\widehat{A}'v^* - (\overline{f}(\theta + v^*) - \overline{f}(\theta))]_{i^*}$$
$$\le \|\widehat{A} - \widehat{A}'\|_1 \|v^*\|_1$$

Therefore, $\text{dist}_{\mathcal{N}}(\widehat{A}, \theta)$ is Lipschitz continuous in $\widehat{A}$. The same argument implies the Lipschitz continuity of $\text{dist}_{\mathcal{N}}(\widehat{A}, \theta)$ in $\theta$.

For (iii), the first claim follows from the definition of $\mathcal{A}(\theta)$ in (26). By the definition of directional derivative,

$$\overline{f}'(\theta; v) = \overline{f}(\theta + v) - \overline{f}(\theta) + o(\|v\|) \tag{73}$$

where $o(s)/s \to 0$ as $s \downarrow 0$. Given $\text{dist}_{\mathcal{N}}(\widehat{A}, \theta) = 0$, we have for each $v \in \mathbb{R}^d$,

$$\widehat{A}v \le \overline{f}(\theta + v) - \overline{f}(\theta) + b_T \|v\|^2 \mathbf{1} = \overline{f}'(\theta; v) + o(\|v\|)$$
$$-\widehat{A}v \le \overline{f}(\theta - v) - \overline{f}(\theta) + b_T \|v\|^2 \mathbf{1} = \overline{f}'(\theta; -v) + o(\|v\|)$$

Using $\overline{f}'(\theta; -v) = -\overline{f}'(\theta; v)$ gives

$$\overline{f}'(\theta; v) - o(\|v\|) \le \widehat{A}v \le \overline{f}'(\theta; v) + o(\|v\|)$$

With $\overline{f}'(\theta; sv)/s = \overline{f}'(\theta; v)$ for $s > 0$, replace $v$ by $sv$ in the above inequality and divide:

$$\overline{f}'(\theta; v) - \frac{o(s\|v\|)}{s} \le \widehat{A}v \le \overline{f}'(\theta; v) + \frac{o(s\|v\|)}{s}$$

Letting $s \downarrow 0$ gives $\widehat{A}v = \overline{f}'(\theta; v)$. $\qquad\square$

**Proposition A.18.** *Under Assumptions (A1)-(A2) and* (54)*,*

(i) *The component-wise inequality holds:*

$$\widehat{A}_n v \le \overline{f}(\theta_n + v) - \overline{f}(\theta_n) + b_T \|v\|^2 \mathbf{1} + o(1), \qquad n \to \infty \tag{74}$$

*where $o(1) \to 0$ as $n \to \infty$ uniformly in $\|v\| \le 1$.*

(ii) $\lim_{n \to \infty} \text{dist}_{\mathcal{N}}(\widehat{A}_n, \theta_n) = 0$ *a.s..*

(iii) *Let $\{\theta_{n_k}\}$ be a subsequence of $\{\theta_n\}$ that converges to some $\theta^\circ \in \mathbb{R}^d$ a.s.. Then,*

$$\lim_{k\to\infty} \, dist(\widehat{A}_{n_k}, \mathcal{A}(\theta^\circ)) = 0\,, \qquad a.s. \tag{75}$$

*where $dist(\widehat{A}_{n_k}, \mathcal{A}(\theta^\circ))$ denotes the Euclidean distance between $\widehat{A}_{n_k}$ and the set $\mathcal{A}(\theta^\circ)$.*

*Proof.* For fixed $n$ and $v \in \mathbb{R}^d$, let $\mathcal{U} : [t_{m(n)}, t_n] \to \mathbb{R}^d$ denote the solution of the following linear integral equation

$$\mathcal{U}_t = \mathcal{U}_{t_{m(n)}} - \int_{t_{m(n)}}^{t} \mathcal{U}_\tau \, d\tau + (t - t_{m(n)})[\overline{f}(\theta_n+v) - \overline{f}(\theta_n) + b_T\|v\|^2]\,, \qquad \mathcal{U}_{t_{m(n)}} = \bar{\mathcal{A}}_{t_{m(n)}} v \tag{76}$$

With $n$ fixed, $\delta_n \triangleq \max_i \big|[o(1)]_i\big|$ in (68) can be viewed as a positive constant. We claim that $\bar{\mathcal{A}}_{t_n} v \le \mathcal{U}_{t_n} + \mathbf{1}\delta_n$. Suppose the claim is not true. Then $[\bar{\mathcal{A}}_{t_n} v]_i > [\mathcal{U}_{t_n}]_i + \delta_n$ for some index $i$ between 1 and $d$. Because $\bar{\mathcal{A}}_t v$ and $\mathcal{U}_t$ are both continuous functions over $[t_{m(n)}, t_n]$ and $\bar{\mathcal{A}}_{t_{m(n)}} v = \mathcal{U}_{t_{m(n)}}$, there exists $t \in [t_{m(n)}, t_n)$ such that $[\bar{\mathcal{A}}_t \, v]_i = [\mathcal{U}_t]_i$ and $[\bar{\mathcal{A}}_\tau v]_i > [\mathcal{U}_\tau]_i$ for $\tau \in (t, t_n)$. Consequently, combing (68) and (76) gives

$$\delta_n < [\bar{\mathcal{A}}_{t_n} v - \mathcal{U}_{t_n}]_i \le [\bar{\mathcal{A}}_t \, v - \mathcal{U}_t]_i - \int_t^{t_n} [\bar{\mathcal{A}}_\tau v - \mathcal{U}_\tau]_i \, d\tau + \delta_n < \delta_n$$

which is a contradiction. Therefore, $\bar{\mathcal{A}}_{t_n} v \le \mathcal{U}_{t_n} + \mathbf{1}\delta_n$.

The integral equation (76) has the solution,

$$\mathcal{U}_t = \exp(t_{m(n)} - t)\mathcal{U}_{t_{m(n)}} + (1 - \exp(t_{m(n)} - t))[\overline{f}(\theta_n+v) - \overline{f}(\theta_n) + b_T\|v\|^2]\,, \qquad t \in [t_{m(n)}, t_n]$$

Consequently,

$$\widehat{A}_n v \le \overline{f}(\theta_n + v) - \overline{f}(\theta_n) + b_T\|v\|^2 \mathbf{1} + \delta_n \mathbf{1}$$
$$+ \exp(t_{m(n)} - t_n)\big[\mathcal{U}_{t_{m(n)}} - (\overline{f}(\theta_n + v) - \overline{f}(\theta_n) + b_T\|v\|^2)\big]$$

By Lemma A.15 (i), we have $t_{m(n)} - t_n < -\ln(n) + 1$ and hence $\exp(t_{m(n)} - t_n) < e/n$. Therefore,

$$\big\|\exp(t_{m(n)} - t_n)\big[\mathcal{U}_{t_{m(n)}} - [\overline{f}(\theta_n + v) - \overline{f}(\theta_n) + b_T\|v\|^2 \mathbf{1}]\big]\big\| \le \frac{e}{n}[b_{\mathcal{A}} + b_L + b_T]\|v\|$$
$$\le \frac{e}{n}[b_{\mathcal{A}} + b_L + b_T]$$

which goes to zero as $n \to \infty$. This proves (i), and (ii) follows by the definition of $dist_{\mathcal{N}}$.

We prove (iii) by contradiction: Suppose (75) does not hold. Then there exists a constant $\delta > 0$ and a subsequence $\{\widehat{A}_{n_k}\}$ such that $dist(\widehat{A}_{n_k}, \mathcal{A}(\theta^\circ)) \ge \delta$ for each $k$. Without loss of generality, the subsequence is convergent, with limit $\widehat{A}^\circ$ satisfying $dist(\widehat{A}^\circ, \mathcal{A}(\theta^\circ)) \ge \delta$. However, combining statement (i) and Prop. A.17 (iii) gives

$$dist(\widehat{A}^\circ, \mathcal{A}(\theta^\circ)) = 0$$

which is a contradiction. $\qquad\square$

### A.5.3 Proofs of Prop. A.11 and Prop. A.12

In this subsection, the time processes involved all refer to those defined in Section A.5.1 with respect to the slow time scale (55).

*Proof of Prop. A.11.* The Lipschitz continuity of $\{\bar{w}_t\}$ and $\{\bar{c}_t\}$ follows directly from boundedness of $\{\theta_n\}$.

At a point of differentiability, let $v_t = \frac{d}{dt}\bar{w}_t = \mathcal{G}_t f(\theta_{[t]}, \Phi_{[t]+1})$ and recall that $\sup_t \|v_t\| \le b_f$. Whenever exists, the derivative of $\bar{c}_t$ may be represented as the directional derivative of $\overline{f}(\bar{w}_t)$ along direction $v_t$:

$$\frac{d}{dt}\bar{c}_t = \lim_{s\to 0} \frac{\overline{f}(\bar{w}_{t+s}) - \overline{f}(\bar{w}_t)}{s} = \lim_{s\downarrow 0} \frac{\overline{f}(\bar{w}_{t+s}) - \overline{f}(\bar{w}_t)}{s} = \overline{f}'(\bar{w}_t; v_t)$$
$$= \lim_{s\uparrow 0} \frac{\overline{f}(\bar{w}_{t+s}) - \overline{f}(\bar{w}_t)}{s} = -\overline{f}'(\bar{w}_t; -v_t) \tag{77}$$

Prop. A.17 (ii) combined with Prop. A.18 (ii) gives

$$\lim_{t\to\infty} \text{dist}_{\mathcal{N}}(\bar{\mathcal{A}}_t, \bar{w}_t) \leq 0\,, \qquad a.s. \tag{78}$$

Let $\eta_t := \max(1/t, \text{dist}_{\mathcal{N}}(\bar{\mathcal{A}}_t, \bar{w}_t))$, satisfying $\eta_t > 0$ and $\eta_t \to 0$ as $t \to \infty$. There exists $T_\bullet < \infty$ *a.s.* such for $t \geq T_\bullet$,

$$\begin{aligned}
\bar{\mathcal{A}}_t v_t - \overline{f}'(\bar{w}; v_t) &= \frac{1}{\sqrt{\eta_t}} \big[\bar{\mathcal{A}}_t \sqrt{\eta_t} v_t - \overline{f}'(\bar{w}_t; \sqrt{\eta_t} v_t)\big] \\
&= \frac{1}{\sqrt{\eta_t}} \big[\bar{\mathcal{A}}_t \sqrt{\eta_t} v_t - [\overline{f}(\bar{w}_t + \sqrt{\eta_t} v_t) - \overline{f}(\bar{w}_t)]\big] + o(\|v_t\|) \\
&\leq (1 + b_T b_f) \sqrt{\eta_t} \mathbf{1} + o(\|v_t\|)
\end{aligned} \tag{79}$$

where the second equality follows from (77) and the last inequality holds given $\text{dist}_{\mathcal{N}}(\bar{\mathcal{A}}_t, \bar{w}_t) \leq \eta_t$ and $\|v_t\|$ is uniformly bounded by $b_f$.

At points of differentiability, we apply $\overline{f}'(\bar{w}_t; v_t) = -\overline{f}'(\bar{w}_t; -v_t)$ from (77):

$$-\bar{\mathcal{A}}_t v_t + \overline{f}'(\bar{w}; v_t) \leq (1 + b_T b_f) \sqrt{\eta_t} \mathbf{1} + o(\|v_t\|)$$

Consequently,

$$\|\bar{\mathcal{A}}_t \tfrac{d}{dt} \bar{w}_t - \tfrac{d}{dt} \bar{c}_t\|_\infty \leq (1 + b_T b_f) \sqrt{\eta_t} + o(\|v_t\|)$$

where $\|\cdot\|_\infty$ denotes the infinity norm. The right hand side of above inequality is bounded and converges to zero as $t \to \infty$. Since the derivatives of $\bar{w}_t$ and $\bar{c}_t$ exist *a.e.*, we have for each $T > 0$,

$$\int_{T_0}^{T_0+T} \|\bar{\mathcal{A}}_t \tfrac{d}{dt} \bar{w}_t - \tfrac{d}{dt} \bar{c}_t\|_\infty \, dt \leq \int_{T_0}^{T_0+T} (1 + b_T b_f) \sqrt{\eta_t} + o(\|v_t\|) \, dt$$

The desired result follows from Dominated Convergence Theorem.

Part (ii) is obtained from (i):

$$\begin{aligned}
\bar{c}_{T_0+t} &= \bar{c}_{T_0} + \int_{T_0}^{T_0+t} \frac{d}{d\tau} \bar{c}_\tau \, d\tau \\
&= \bar{c}_{T_0} + \int_{T_0}^{T_0+t} \bar{\mathcal{A}}_\tau \bar{\mathcal{G}}_\tau f(\theta_{[\tau]}, \Phi_{[\tau]+1}) \, d\tau + o(1)\,, \qquad T_0 \to \infty \\
&= \bar{c}_{T_0} + \int_{T_0}^{T_0+t} \bar{\mathcal{A}}_\tau \bar{\mathcal{G}}_\tau \overline{f}(\bar{w}_\tau) \, d\tau + o(1)\,, \qquad T_0 \to \infty
\end{aligned}$$

where the last equality follows from standard ODE arguments for stochastic approximation [5].

For (iii), $\|\bar{c}_t\|^2$ is Lipschitz continuous in $t$ given boundedness of $\{\theta_n\}$. Hence by the same argument in (ii),

$$\begin{aligned}
\|\bar{c}_{T_0+t}\|^2 &= \|\bar{c}_{T_0}\|^2 + 2\int_{T_0}^{T_0+t} \bar{c}_\tau^{\mathsf{T}} \frac{d}{d\tau} \bar{c}_\tau \, d\tau \\
&= \|\bar{c}_{T_0}\|^2 + 2\int_{T_0}^{T_0+t} \bar{c}_\tau^{\mathsf{T}} \bar{\mathcal{A}}_\tau \bar{\mathcal{G}}_\tau \bar{c}_\tau \, d\tau + o(1)\,, \qquad T_0 \to \infty
\end{aligned}$$

$\square$

*Proof of Prop. A.12.* (i) follows from the Lipschitz continuity of $\overline{f}$.

Let $\{T_k\}$ be a sequence such that $\overline{\Gamma}^{T_k} \to \Gamma$ for each of the four components: $\overline{\Gamma}_i^{T_k}$, $1 \leq i \leq 4$. Since $\overline{\Gamma}_3^{T_k} \to \mathcal{A}$ weakly in $L_2([0,T]; \mathbb{R}^{d\times d})$ as $k \to \infty$, by the Banach-Saks theorem, there exists a subsequence $\{T_{n_k}\}$ such that

$$\frac{1}{N}\sum_{k=1}^{N} \overline{\Gamma}_3^{T_{n_k}}(t) \to \mathcal{A}_t\,, \qquad a.e.\, t \in [0,T]\,, \qquad N \to \infty$$

Without loss of generality, we can modifying $\mathcal{A}_t$ on a Lebesgue-null set such that the convergence above is pointwise. We also have $\overline{\Gamma}_1^{T_{n_k}}(t) \to w_t$ as $k \to \infty$ for each $t \in [0, T]$. By Prop. A.18 (ii),

$$\lim_{k \to \infty} \operatorname{dist}(\overline{\Gamma}_3^{T_{n_k}}(t), \mathcal{A}(w_t)) = 0, \qquad t \in [0, T]$$

It follows from definition (26) that the set $\mathcal{A}(\theta)$ is convex for each $\theta$. Then,

$$\lim_{N \to \infty} \operatorname{dist}(\frac{1}{N} \sum_{k=1}^{N} \overline{\Gamma}_3^{T_{n_k}}(t), \mathcal{A}(w_t)) = 0, \qquad t \in [0, T]$$

Therefore, $\mathcal{A}_t \in \mathcal{A}(w_t)$ for each $t \in [0, T]$. This proves (ii).

Given that $\overline{\Gamma}_4^{T_0}$ is positive semi-definite pointwise and uniformly bounded, the same arguments establish (iii).

Since $\Gamma_2^{T_k} \to c$ uniformly over $[0, T]$ and $\Gamma_4^{T_k} \to \mathcal{H}$ weakly, $\overline{\Gamma}_4^{T_k} \overline{\Gamma}_2^{T_k}$ converges to $\mathcal{H}c : [0, T] \to \mathbb{R}^d$ weakly. The ODE (57a) follows from Prop. A.11 (ii). For (57b), since $b_\lambda := \sup_n \lambda_{\max}(\widehat{A}_n^\intercal \widehat{A}_n)$ is finite,

$$-[\varepsilon I + \widehat{A}_n^\intercal \widehat{A}_n]^{-1} \le -\frac{1}{\varepsilon + b_\lambda} I, \qquad n \ge 1$$

Combining this inequality with Prop. A.11 (iii) implies

$$\|\bar{c}_{T_0+t}\|^2 \le \|\bar{c}_{T_0}\|^2 - \frac{2}{\varepsilon + b_\lambda} \int_{T_0}^{T_0+t} \|\bar{\mathcal{A}}_\tau^\intercal \bar{c}_\tau\|^2 \, d\tau + o(1), \qquad T_0 \to 0 \tag{80}$$

We can show that $\{(\overline{\Gamma}_3^{T_k})^\intercal \overline{\Gamma}_2^{T_k}\}$ converges weakly to $\mathcal{A}^\intercal c$ in $L_2([0, T]; \mathbb{R}^d)$ by the sames arguments that we used to establish $\overline{\Gamma}_4^{T_k} \overline{\Gamma}_2^{T_k} \to \mathcal{H}c$ weakly. Applying [17, Theorem 2.2.1], we obtain for each $t \in [0, T]$,

$$\int_0^t \|\mathcal{A}_\tau^\intercal c_\tau\|^2 d\tau \le \liminf_{k \to \infty} \int_0^t \|[\overline{\Gamma}_3^{T_k}(\tau)]^\intercal \overline{\Gamma}_2^{T_k}(\tau)\|^2 \, d\tau$$

Consequently,

$$\|c_t\|^2 \le \|c_0\|^2 - \frac{2}{\varepsilon + b_\lambda} \int_0^t \|\mathcal{A}_\tau^\intercal c_\tau\|^2 d\tau$$

$\square$

### A.5.4 General eligibility vector $\zeta$

We finally come to the general model in which (54) is relaxed. For the sake of analysis, the two functions $\mathcal{D}, \zeta$ in (19b) are assumed to be parameterized by separate parameters $\theta, \xi \in \mathbb{R}^d$: $\mathcal{D}(\theta, z), \zeta(\xi, x, u)$. This is only for clarifying calculations – in the end we do impose $\theta = \xi$. Decompose the function $\zeta : \mathbb{R}^d \times \mathsf{X} \times \mathsf{U} \to \mathbb{R}^d$ into its positive and negative components: $\zeta = \zeta^+ - \zeta^-$, with $\zeta^+ = \max(\zeta, 0)$ and $\zeta^- = \max(-\zeta, 0)$. Define functions $f^+, f^- : \mathbb{R}^d \times \mathbb{R}^d \times \mathsf{Z} \to \mathbb{R}^d$ by

$$f^+(\xi, \theta, z) = \zeta^+(\xi, x, u) \mathcal{D}(\theta, z), \qquad f^-(\xi, \theta, z) = \zeta^-(\xi, x, u) \mathcal{D}(\theta, z)$$

Next define functions $\overline{f}^+, \overline{f}^- : \mathbb{R}^d \times \mathbb{R}^d \to \mathbb{R}^d$ by

$$\overline{f}^+(\xi, \theta) = \mathsf{E}_\varpi[f^+(\xi, \theta, \Phi_{n+1})], \qquad \overline{f}^-(\xi, \theta) = \mathsf{E}_\varpi[f^-(\xi, \theta, \Phi_{n+1})]$$

Let $\mathcal{A}^+(\theta), \mathcal{A}^-(\theta)$ denote the sets of generalized subgradients of $\overline{f}^+, \overline{f}^-$ with respect to $\theta$ based on (26). Explicit representations of $\mathcal{A}^+(\theta)$ and $\mathcal{A}^-(\theta)$ can be obtained as in Lemma A.13. With general eligibility vector $\zeta$, let $\mathcal{A}(\theta)$ denote the set

$$\mathcal{A}(\theta) := \{A^+ - A^- + \mathsf{E}_\varpi[\mathcal{D}(\theta, \Phi_{n+1}) \partial_\xi \zeta_n(\theta)] : A^+ \in \mathcal{A}^+(\theta), A^- \in \mathcal{A}^-(\theta)\} \tag{81}$$

At each $\theta \in \mathbb{R}^d$, denote

$$\overline{f}^+(\theta; v) := \lim_{s \downarrow 0} \frac{\overline{f}^+(\theta, \theta + sv) - \overline{f}^+(\theta, \theta)}{s}$$

with $\overline{f}^-(\theta;v)$ is defined similarly. Then the directional derivative $\overline{f}'(\theta;v)$ can be expressed

$$\overline{f}'(\theta;v) = \lim_{s \downarrow 0} \frac{\overline{f}(\theta + sv) - \overline{f}(\theta)}{s} = \overline{f}^+(\theta;v) - \overline{f}^-(\theta;v) + \mathsf{E}_\varpi[\mathcal{D}(\theta,\Phi_{n+1})\partial_\xi \zeta_n]v, \quad \theta,v \in \mathbb{R}^d$$
(82)

Decompose $A_{n+1}$ in (21a) as $A_{n+1} = A_{n+1}^+ - A_{n+1}^- + A_{n+1}^\zeta$:

$$A_{n+1}^+ = \zeta_n^+[\gamma \partial_\theta Q^\theta(X_{n+1}, \phi^{\theta_n}(X_{n+1})) - \partial_\theta Q^\theta(X_n, U_n)]$$
$$A_{n+1}^- = \zeta_n^-[\gamma \partial_\theta Q^\theta(X_{n+1}, \phi^{\theta_n}(X_{n+1})) - \partial_\theta Q^\theta(X_n, U_n)]$$
$$A_{n+1}^\zeta = \mathcal{D}(\theta_n, \Phi_{n+1})\partial_\xi \zeta_n$$

Accordingly, the matrix gain is decomposed: $\widehat{A}_{n+1} = \widehat{A}_{n+1}^+ - \widehat{A}_{n+1}^- + \widehat{A}_{n+1}^\zeta$, and each component can be expressed in the recursive form:

$$\widehat{A}_{n+1}^+ = \widehat{A}_n^+ + \beta_{n+1}[A_{n+1}^+ - \widehat{A}_n^+]$$
$$\widehat{A}_{n+1}^- = \widehat{A}_n^- + \beta_{n+1}[A_{n+1}^- - \widehat{A}_n^-]$$
$$\widehat{A}_{n+1}^\zeta = \widehat{A}_n^\zeta + \beta_{n+1}[A_{n+1}^\zeta - \widehat{A}_n^\zeta]$$

**Analysis of $\{\widehat{A}_n^+, \widehat{A}_n^-, \widehat{A}_n^\zeta\}$ over the fast time scale:** Consider the fast time scale defined by (63). The conclusions in Section A.5.2 hold for each of $\{\widehat{A}_n^+\}$ and $\{\widehat{A}_n^-\}$. While $\{\widehat{A}_n^\zeta\}$ can be treated using standard SA arguments since $A_{n+1}^\zeta$ is Lipschitz continuous with respect to $\theta_n$ under (A2). We obtain an extension of Prop. A.18:

**Proposition A.19.** *The following hold:*

(i) *As $n \to \infty$,*

$$\widehat{A}_n^+ v \le \overline{f}^+(\theta_n, \theta_n + v) - \overline{f}^+(\theta_n, \theta_n) + b_T \|v\|^2 \mathbf{1} + o(1)$$
$$\widehat{A}_n^- v \le \overline{f}^-(\theta_n, \theta_n + v) - \overline{f}^-(\theta_n, \theta_n) + b_T \|v\|^2 \mathbf{1} + o(1)$$

*where $o(1) \to 0$ as $n \to \infty$, uniformly in $\|v\| \le 1$.*

(ii) *Let $\{\theta_{n_k}\}$ be a subsequence of $\{\theta_n\}$ that converges to some $\theta^\circ \in \mathbb{R}^d$ a.s.. Then,*

$$\lim_{k\to\infty} dist(\widehat{A}_{n_k}^+, \mathcal{A}^+(\theta^\circ)) = 0, \qquad \lim_{k\to\infty} dist(\widehat{A}_{n_k}^-, \mathcal{A}^-(\theta^\circ)) = 0, \qquad a.s.$$

(iii) $\widehat{A}_n^\zeta = \mathsf{E}_\varpi[\mathcal{D}(\theta_n, \Phi_{n+1})\partial_\xi \zeta_n] + o(1)$.

**Analysis of $\{\theta_n\}$ over the slow time scale:** Going back to the slow time scale defined by (55), define the continuous time processes $\{\bar{w}_t, \bar{c}_t : t \ge 0\}$ as before. Define similarly the piecewise constant time processes $\{\bar{\mathcal{A}}_t, \bar{\mathcal{G}}_t : t \ge 0\}$ as well as the three components $\{\bar{\mathcal{A}}_t^+, \bar{\mathcal{A}}_t^-, \bar{\mathcal{A}}_t^\zeta : t \ge 0\}$.

**Proposition A.20.** *The conclusions of Prop. A.11 and Prop. A.12 hold for general eligibility vectors, subject to the modified definition of $\mathcal{A}(\theta)$ in (81).*

*Proof.* For the three claims of Prop. A.11, it suffices to prove that Prop. A.11 (i) holds with the new definition (81) of $\mathcal{A}(\theta)$. The rest of the claims then follow from (i).

At a point $t$ where both $\bar{w}_t$ and $\bar{c}_t$ are differentiable, denote $v_t = \frac{d}{dt}\bar{w}_t$. Consider

$$\lim_{s\to 0} \frac{\overline{f}^+(\bar{w}_t, \bar{w}_{t+s}) - \overline{f}^+(\bar{w}_t, \bar{w}_t)}{s} = \lim_{s\to 0} \sum_{x,u} \varpi(x,u)\zeta^+(\bar{w}_t, x, u)\frac{\varsigma^{\bar{w}_{t+s}}(x,u) - \varsigma^{\bar{w}_t}(x,u)}{s}$$

By Lemma A.8, $\varsigma^{\bar{w}_t}(x,u)$ is differentiable for each state-action pair and *a.e.* $t$, and hence

$$\overline{f}^+(\bar{w}_t; v_t) = -\overline{f}^+(\bar{w}_t; -v_t), \qquad \text{for } a.e.\, t \in \mathbb{R}_+$$

The same arguments imply $\overline{f}^-(\bar{w}_t; v_t) = -\overline{f}^-(\bar{w}_t; -v_t)$ for *a.e.* $t \in \mathbb{R}_+$. Then, with Prop. A.19 (i), the same arguments used to establish Prop. A.11 (i) yield those conclusions: For each $T > 0$,

$$\lim_{T_0 \to \infty} \int_{T_0}^{T_0+T} \|\bar{\mathcal{A}}_t^+ v_t - \overline{f}^+(\bar{w}_t; v_t)\|_\infty \, dt = 0$$

$$\lim_{T_0 \to \infty} \int_{T_0}^{T_0+T} \|\bar{\mathcal{A}}_t^- v_t - \overline{f}^-(\bar{w}_t; v_t)\|_\infty \, dt = 0$$

It follows from (82) that

$$\tfrac{d}{dt} \bar{c}_t = \overline{f}^+(\bar{w}_t; v_t) - \overline{f}^-(\bar{w}_t; v_t) + \mathsf{E}_\varpi[\mathcal{D}(\bar{w}_t, \Phi_{n+1}) \partial_\xi \zeta_n] v_t$$

Therefore,

$$\int_{T_0}^{T_0+T} \|\bar{\mathcal{A}}_t v_t - \tfrac{d}{dt} \bar{c}_t\|_\infty \, dt \le \int_{T_0}^{T_0+T} \|\bar{\mathcal{A}}_t^+ v_t - \overline{f}^+(\bar{w}_t; v_t)\|_\infty + \|\bar{\mathcal{A}}_t^- v_t - \overline{f}^-(\bar{w}_t; v_t)\|_\infty \, dt$$

$$+ \int_{T_0}^{T_0+T} \|\bar{\mathcal{A}}_t^\zeta v_t - \mathsf{E}_\varpi[\mathcal{D}(\bar{w}_t, \Phi_{n+1}) \partial_\xi \zeta_n] v_t\|_\infty \, dt$$

where the right hand side of the above inequality goes to 0 as $T_0 \to \infty$.

For the conclusions of Prop. A.12, we only need to prove (ii) with the new $\mathcal{A}(\theta)$. Let $\mathcal{A}_t^+, \mathcal{A}_t^-, \mathcal{A}_t^\zeta$ denote the weak sub-sequential limits of $\{\bar{\mathcal{A}}_{T_0+t}^+, \bar{\mathcal{A}}_{T_0+t}^-, \bar{\mathcal{A}}_{T_0+t}^\zeta : T_0 \ge 0, 0 \le t \le T\}$ respectively. By Prop. A.19 (ii), the same arguments used for Prop. A.12 (ii) apply to each of $\mathcal{A}_t^+$ and $\mathcal{A}_t^-$,

$$\mathcal{A}_t^+ \in \mathcal{A}^+(w_t), \qquad \mathcal{A}_t^- \in \mathcal{A}^-(w_t), \qquad t \in [0, T]$$

We also have $\mathcal{A}_t^\zeta = \mathsf{E}_\varpi[\mathcal{D}(w_t, \Phi_{n+1}) \partial_\xi \zeta_n]$ from Prop. A.19 (iii). Therefore, $\mathcal{A}_t = \mathcal{A}_t^+ - \mathcal{A}_t^- + \mathcal{A}_t^\zeta$, and $\mathcal{A}_t \in \mathcal{A}(w_t)$ for each $t \in [0, T]$. $\qquad\square$

Following the same arguments as in Section A.5.1, the ODE approximations and ODE limits established in Prop. A.20 imply the following extension of Thm. 2.1:

**Theorem A.21.** *The conclusions of Thm. 2.1 hold, subject to the modified definition of $\mathcal{A}(\theta)$ in (81).*

### A.6 Numerical Results: Implementation details

**Complexity of Zap Q-learning**  For the Zap Q-learning algorithm (21), per-iteration complexity comes from various sources:

 (i) Computation of $f(\theta_n, \Phi_{n+1})$ involves a maximum to obtain $\underline{Q}^{\theta_n}$ in (19a).

 (ii) The derivatives $A_{n+1} = \partial_\theta f(\theta_n, \Phi_{n+1})$ are easily computed for linear parameterization of $Q^\theta$, but require back-propagation in a neural network function approximation architecture.

 (iii) Computation of $G_{n+1} f(\theta_n, \Phi_{n+1})$ in (21c) and (21d) requires (i) multiplication of two $d \times d$ matrices, and (ii) multiplying a matrix inverse and a vector. Each of these two steps has worst case computational complexity $O(d^3)$.

As discussed in Section 3, the complexity in (iii) can be reduced by updating the gain only periodically, while continuously updating estimates of $A(\theta_n)$.

The complexity bound $O(Nd^3/N_d + Nd^2)$ given in Section 3 is based on gain updates performed only at integer multiples of $N_d$. This bound is based on the accounting (i)—(iii) above: $O(d^2)$ complexity per iteration in (21b), and $O(d^3)$ complexity for the matrix inverse (as well as the product $\hat{A}_{n+1}^\intercal \hat{A}_{n+1}$ appearing in (21c)).

**Meta-parameters in experiments**  We used $\varepsilon = 10^{-6}$ in (21c) for Mountain car and Acrobot, $\varepsilon = 10^{-4}$ for Cartpole.

For the decreasing step-size rule, we used $\rho = 0.85$ and $n_0 = 100$ in (22). For constant step-size experiments, we used

$$\alpha_n \equiv \alpha, \qquad \beta_n \equiv \beta = 100\alpha$$

The choice of $\alpha$ itself was problem specific: $\alpha = 0.002$ for the network of size $6 \times 3$ in the Mountain car example; $\alpha = 0.005$ for other experiments using constant step-size. The average reward $\mathcal{R}(\phi^{\theta_n})$ defined in (31) was estimated by running 100 independent simulations following the policy $\phi^{\theta_n}$. The deterministic upper bound $\bar{\tau}$ was 200 for Mountain car and Acrobot, and $\bar{\tau} = 1000$ for Cartpole.

**Q-network**    The input space $\mathsf{U}$ in each of the examples is a finite set of scalars. Recall that the size of neural networks indicated in Figure 1 refers to the size of hidden layers, with the input to the network $(x, u)$ and the output $Q^{\theta}(x, u)$; hence, in the Cartpole example with $(x, u) \in \mathbb{R}^5$, the network size $30 \times 24 \times 16$ corresponds to $\theta \in \mathbb{R}^d$, with $d = 1341$:

$$d = (5 + 1) * 30 + (30 + 1) * 24 + (24 + 1) * 16 + (16 + 1) = 1341$$

where each $+\,1$ accounts for a bias parameter.

**Policy**    The theory developed in this paper assumes a randomized stationary policy for exploration. In our experiments, we apply the parameter-dependent $\epsilon$-greedy exploration: At iteration $n$,

$$U_n = \begin{cases} \phi^{\theta_n}(X_n), & \text{with probability } 1 - \epsilon \\ \texttt{rand}, & \text{with probability } \epsilon \end{cases}$$

We set $\epsilon = 0.4$ for the Mountain Car and Acrobot, and $\epsilon = 0.2$ for Cartpole.