[Reviews · NeurIPS 2020]

Review 1

Summary and Contributions: This paper introduces a version of Zap Q-learning that can be applied to arbitrary approximation architectures for Q-functions. Convergence analysis is undertaken, and a version of the algorithm with MLP function approximators is applied to several classical control tasks. POST-REBUTTAL ------------------------ I thank the authors for their response. I appreciate the comments around reorganisation of material, and clarification of some of the technical points I raised. There are two main concerns that I have with the paper that prevent me from strongly recommending acceptance, described below. THEORY Further description of when the required conditions/assumptions for the main technical results are likely to hold would strengthen the paper a lot. One particular point that arose during discussion with other reviewers was the assumption of boundedness of (\theta_n)_n. If we assume boundedness of parameter sequences, then a lot of the difficulties that standard Q-learning experiences also disappear (for example, the classic examples by Tsitsiklis and Van Roy (https://www.mit.edu/~jnt/Papers/J063-97-bvr-td.pdf) are ruled out). If the authors can demonstrate small examples such as those of this paper where Zap SA converges but a standard Q-learning algorithm still has parameter divergence, I think this would go a long way to showing the reasonableness of the assumptions. EXPERIMENTS I wasn't convinced by the additional experiments comparing against DQN in the rebuttal. The primary reason is that judging by Appendix A.6, some hyperparameter sweeping was been carried out for the Zap SA experiments (I'd encourage you to describe how this was carried out), whereas I understood the DQN experiments to have been carried out with default hyperparameters (although with the modified, smaller network topology). I think to be able to make a meaningful comparison against DQN, similar hyperparameter sweeps should be undertaken for both methods -- it's not surprising that the method without any hyperparameter sweeping performed worse. Other comments: As a minor note, I didn't fully understand the comment near the top of the rebuttal about achieving Lipschitz parametrisation by projecting into a compact (convex) region of parameter space -- I understand that this guarantees a Lipschitz approximation as long as the approximator is continuously differentiable, but thought that an additional projection step such as this would lead to a non-trivial change in the analysis of the algorithm? The additional experiments regarding the frequency of updating \hat{A}_n seem to indicate an interesting trade-off between stability (frequent updating of \hat{A}_n) vs. lower computational cost (infrequent updating). This seems like a nice phenomenon to explore further, particularly if the frequency can be set adaptively.

Strengths: The core algorithm introduced here is interesting and novel, and the technical quality of the work is high; guarantees for RL algorithms with non-linear function approximation are difficult to come by. The experiments indicate that the main algorithm presented is compatible with neural network function approximation, and therefore these ideas ought to be of interest to the deep RL community.

Weaknesses: Assumptions. The assumptions required for the convergence analysis are quite restrictive, and therefore are unlikely to be applicable in practical scenarios. The most salient of these are the need for a single data-generating policy to be used throughout training (meaning that approaches like epsilon-greedy, or regret-minimising policies, cannot be used) and the requirement of Lipschitz function approximators (ruling out most networks using ReLU activations, for example). However, developing theory for general non-linear function approximation is extremely difficult, and I don't think the theory has to be universally applicable for acceptance. Experiments. The experiments section is somewhat limited, and whilst it demonstrates that the proposed algorithm is empirically compatible with neural network function approximation, it does not shed any light on how this approach compares to established methods for stabilising training in non-linear regimes. Presentation. I would have preferred a different balance of material in the main paper. I found the first four pages to be a fairly wandering review of a wide range of topics in stochastic approximation, but without really describing any specifics to reinforcement learning in much detail (temporal differences aren't defined until p5, for example), and perhaps a more focussed review of Q-learning and Zap Q-learning would make the thread easier to follow, and leave more room for in-depth discussion of the main algorithm and its analysis later in the paper.

Correctness: The technical quality of the main paper is of high quality, and I did not find any errors here. I have not been able to check the 20 page appendix line-by-line due to time constraints in the reviewing period, and so have made spot checks of several parts of the appendix. Some minor comments are given in more detail below, but I believe these to be correct.

Clarity: I found the writing clear and precise, and the technical content to be of a higher quality than is typical for NeurIPS papers. The supplementary material could do with rearranging, or perhaps an additional section at the beginning that explains what the various sections pertain to in the main paper.

Relation to Prior Work: The paper contextualises itself well against earlier work in stochastic approximation and analysis of RL algorithms.

Reproducibility: Yes

Additional Feedback: Detailed comments are given below. The discussion around Eqns (10) and (11) was difficult to parse. Why does (\Delta_n)_n satisfy a CLT? It is not obvious to me why this should be the case. Do authors mean something like the cumulative sums of this sequence satisfy a CLT? I was also unsure about \Delta, and hence \Phi, being a doubly infinite sequence in Eqn (11). Implicitly it is indexed by the natural numbers in earlier discussion - is there an implicit assumption that actually the Markov chain is in equilibrium at all timesteps? Contributions: Do the authors mean "A generalisation of the Zap Q-learning algorithm…"? "Extensions to other criteria are straightforward" - can the authors clarify whether this means that related problems can straightforwardly be expressed in terms of average cost, finite horizon etc., or that the algorithm and analysis in the paper for the specific case of infinite horizon discounted cost can be straightforwardly applied to these other problems too? "The generalized subgradient of f(\theta, z) exists under additional assumptions". Can the authors make these assumptions explicit? At present, I'm unsure from this paragraph whether the analysis the authors want to undertake will apply to all f obtained from the MDP application of the paper, or whether additional assumptions are needed. I would have found more discussion around the Zap SA algorithm itself useful, such as intuitions for the form of the algorithm, explanations on the connections to the Newton-Raphson method, etc. Assumption A1: I am confused by this assumption: each choice of "policy" (i.e. a manner of selecting U based on X) leads to a different Markov chain, with a potentially different invariant distribution. Can the authors clarify what is meant here? Do they mean that they assume data is generated according to a fixed policy throughout, and that the resulting Markov chain satisfies these assumptions? Assumption A2: The assumption that Q_\theta is Lipschitz in \theta seems quite restrictive, and if I understand correctly, would rule out e.g. two-layer ReLU neural networks (since the output of the network can then contain products \theta_1 \theta_2 of two weights) - can the authors comment on this? Line 196: What is meant by bounded here? That the sequence (\theta_n) is bounded almost surely? Or uniformly bounded almost surely? Regarding Section A1, can the authors confirm whether Assumption A2_\infty is required for Theorem 2.1 to hold? Can the authors also comment on when A2_\infty is likely to apply? The limiting eligibility trace condition on \zeta_\infty in Section A.1 seems to rule out e.g. networks with unbounded activation functions. Section A.3 I couldn't see the notation x^k, u^k defined elsewhere - presumably this is just an enumeration of the set X x U? The argument that H^{½} is irreducible seems to rely on Assumption A1, but this isn't particularly clear since Section A.3 is referenced earlier in the main paper that Assumption A1, and the statement of Proposition A.4 itself in Section A.3 doesn't reference these assumptions.


Review 2

Summary and Contributions: The paper discusses the application of Stochastic Approximation to Q Learning. The proposed approach is subsequently applied to (simple) standard benchmarks from the OpenAI gym.

Strengths: The paper potentially provides interesting theoretical insight into an emerging field in Reinforcement Learning. However, this cannot be verified due to the reviewers's lack of expertise in the area but also the insufficient presentation quality of the paper.

Weaknesses: Stochastic Approximation and related work is not the reviewer's field of expertise, despite having some background in Reinforcement Learning. As such, it is expected that, given the theoretical nature of the paper, the theoretical developments in the paper are not easily accessible. However, the paper does not frame the work sufficiently to be able to put the work into the larger context of Q-learning and clearly differentiate the work from prior works. Furthermore, the experimental results at a first glance seem unimpressive. While the contributions of the paper seem to lie in the theoretical developments, the experiments require a better framing. Along these lines, the conclusions can be greatly improved by drawing clearer conclusions (and maybe reducing the amount of discussion about future work).

Correctness: The overall paper requires prior knowledge in a field that the reviewer is not knowledgeable in and thus the correctness of the paper cannot be verified.

Clarity: As discussed above, the paper requires improvements with regard to clarity and structure. The introduction dives too quickly into details without providing a general setting for the paper.

Relation to Prior Work: The paper could be improved to point out the contributions more clearly both in the introduction as well as in the conclusion.

Reproducibility: Yes

Additional Feedback:


Review 3

Summary and Contributions: The main contribution of this paper is to generalize the so-called Zap Q-learning algorithm for a more general class of stochastic approximation. This algorithm is essentially an application of Newton-Raphson method, which is shown to be stable even under nonlinear function approximation, such as neural networks. The authors indeed show that their algorithms are stable under some technical conditions.

Strengths: This paper provides a nice application of Newton-Raphson method in stochastic approximation and reinfocement learning with theoretical guarantees. The results in this paper provide an approach to tackle with the difficulities of handling function approximation in reinforcement learning. I read the authors responses on my comments and others. In general, I think this is a good theoretical paper. My comments are for general extensions, which may not be obvious to address. I would recommend an acceptance for this paper.

Weaknesses: As mentioned in the paper, one issue of their algorithm is to require computing an inverse of a matrix, which can be very expensive for a large scale problem. This is the trade-off between stablity and computation of Newton-Raphson in general. Another difficulty comes from choosing the two step-sizes in their algorithms, which is quite challenging due to the nature of two-time-scale updates.

Correctness: The analytical and experimental results seem to correct. I tried to check the paper carefully but I may also skip some technical analysis in the appendix.

Clarity: The paper is well-written, although it takes a few rounds of reading to grasp the main ideas.

Relation to Prior Work: yes

Reproducibility: Yes

Additional Feedback:


Review 4

Summary and Contributions: This paper presents an extension of the Zap Q-learning algorithm to the function approximation setting, and provides an analysis of its convergence and stability properties. The principal contribution of the paper is this analysis. The method is then evaluated on a selection of problems from the OpenAI gym suite of environments.

Strengths: The extension of Zap Q-learning to the function approximation setting did not previously exist in the literature; while the proposed algorithm is a straightforward adaptation of its tabular analogue, it nonetheless did not previously exist in the literature. The paper provides theoretical convergence guarantees for the proposed method under some assumptions. The analysis involved in proving this is in my opinion the most significant contribution of the paper. The method appears to be learning in the different environments it is evaluated on. The paper area is of great interest to the RL community; the poor convergence properties of RL methods with nonlinear function approximation, along with their sensitivity to hyperparameters, are a significant challenge in applying RL methods to problems of interest. More robust algorithms with better convergence properties could present a significant step forward for the field.

Weaknesses: -The main weakness of the paper is simply that most of its contributions are deferred to the appendix and only alluded to in the main body of the paper; this makes the appendix difficult to navigate, and makes the main body overly abstract so that it’s difficult to understand the mechanics of how the main insights play out. -The modification to zap q-learning is relatively small; it effectively amounts to changing the TD update to be gradient-based and adding a small perturbation inside the matrix inverse (presumably to guarantee invertibility). -The empirical evaluation of the method doesn’t include any comparisons against other baselines. The motivation of zap q-learning lies in its convergence/stability, but this isn’t compared against a baseline such as GQ-learning. I would have liked to see some environments where zap q-learning exhibits better convergence properties than GQ-learning. -Additionally, the method only appears to ‘solve’ (according to the definition given by OpenAI in their benchmarks repository) one of the three tasks. The best methods in acrobot achieve reward closer to -60 and in mountain car a method is only said to have solved the task when it achieves an average return of 110 over 10 episodes. -The analysis is limited to fixed behaviour policies and does not apply to the epsilon-greedy exploration strategies often used in deep RL, or even policy iteration.

Correctness: -I went over the proofs at a high level and didn’t spot any obvious errors, so my main concerns about correctness relate to the assumptions made in the proofs, which I’ll list here. -Assumption A4 seems like it could plausibly be violated for many popular function approximation architectures. -A3 further seems unlikely to be satisfied in practice. One plausible ‘bad minimum’ that would satisfy f(theta_n) = 0 is one corresponding to a layer of ‘dead units’ (one in which all inputs lead to the node outputting zero) in a neural network used as a function approximator. In this case, both the gradient and the output of the network would be zero for all inputs, and so we would obtain an undesirable local optimum of the projected Bellman error. -The requirement that the eligibility vector be twice continuously differentiable likely won’t apply to many function approximation architectures used in practice.

Clarity: I was initially confused by the distinction between contributions (i) and (ii) on page 3. Would it be correct to summarize (i) as being agnostic to the next-action selection rule, and (ii) to refer specifically to the argmax procedure defined in eq (17) ? The structure of the paper makes it difficult to evaluate the significance and generalizability of the theoretical contribution to other methods in reinforcement learning. I would strongly recommend restructuring the paper to give more space to the outline of the proof of the convergence results. As a reader, it was difficult to work through almost 20 pages of dense proofs where in many sections it’s difficult to see the utility of a given proposition when it’s presented. Because much of the utility of a paper like this comes from presenting mathematical tools that can be generalized to other settings in RL, I would like to see the structure of the proof given more space in the paper. My recommendation would be to cut the introduction down to 1-2 pages and to extend the overview of the proof of theorem 2.1.

Relation to Prior Work: -Prior work is well-cited and relevant papers are referred to in the text. Off the top of my head I can't think of any obvious missing citations. The distinction between this paper and previous work is clearly stated.

Reproducibility: Yes

Additional Feedback: -I have a few questions about the empirical evaluation of the method. - Do the environments and function approximation architecture satisfy the assumptions of the convergence results? - Does the \bar{f} term in these environments converge to zero? I’m particularly interested in seeing how \bar{f} compares between GQ and zap Q learning methods with function approximation. -What happens when the matrix inverse term is computed only once at initialization? Does this affect performance at all? What would make me improve my score: improved clarity in the paper with a more explicit discussion of the analysis and intermediate results, and a more rigorous empirical evaluation of the method along the lines discussed in this review. ---------------------------------------------------------------------------------------------- POST-RESPONSE UPDATE I'd first off like to thank the authors for their detailed and thorough response. My concerns re: the non-singularity in (A4) and the roots of \bar{f} have been addressed. Re: epsilon-greedy policies, this does seem like a constraint on the types of settings this method is applicable to, so I will be interested in seeing whether the authors' comment "there is hope that stability can be established, provided we adapt the policy on a slow time scale" pans out in future work. I further appreciate the authors' proposed reorganization of the paper: I think this will greatly clarify the nature of the contributions. I still have two main concerns that are preventing me from confidently updating my score: first, it seems like the Q-learning baseline in the rebuttal used a network that is much smaller than that typically used in DQN agents -- this might have a significant effect on how well the default hyperparameters perform, and so I'd like to see a more fair hyperparameter sweep done on the baseline Q-learner. The experiments are promising but not decisive, and given that the assumptions required for the theory to hold won't necessarily be guaranteed in all deep RL settings, I'd want to see more conclusive evaluations in these settings. I've also given more thought to R1's point regarding the boundedness assumption and share similar concerns.

[Author Response · NeurIPS 2020]

We thank all the reviewers for their time and effort during these tough times. Much of the feedback we received was inspiring, and we have worked day and night to respond. Details and additional numerical results are included below. However, please keep in mind that this is a theory paper. We doubt anyone before now thought it is possible to establish stability of Q-learning with nonlinear function approximation under the general conditions of our paper. The simple examples are included to illustrate the theory, and the stunning reliability of the new algorithms.

**Assumptions (R#1 & R#4):**

**1. Smoothness of the network and $\zeta$ (R#1):**   The Lipschitz condition is only required for the ODE approximation— the Newton-Raphson flow is stable without this condition. Hence more sophisticated numerical methods will likely extend the theory beyond Lipschitz functions. Alternatively, we can project the estimates to a bounded convex region.

**2. On non-singularity in (A4) (R#4):**   Indeed, we find that $A(\theta^*)$ is often singular in neural network architectures. We conjecture that the symmetry of the NN leads to both lack of uniqueness of $\theta^*$, as well as the potential singularity. However: this condition is imposed to obtain a CLT. *It is not crucial for our main results.*

**3. Roots of $\overline{f}$ (R#4):**   We have included fresh experiments below to show that $\overline{f}(\theta_n) \to 0$, and reward plots in the paper are consistent with this conclusion. We have not observed 'bad roots' from 'dead units' in our experiments, but will consider how the algorithm can be modified to avoid this potential threat.

**4. On stability in Section A.1 (R#1):**   *This assumption is not required.* In Theorem 2.1 we *assume* that the parameter sequence is bounded. $A2_\infty$ provides a sufficient condition for boundedness (avoiding projection of parameter estimates).

**5. On $\varepsilon$-greedy policies (R#1):**   Incorporating exploration in these algorithms, while maintaining stability guarantees, is an important topic for future research. We will expand on the discussion currently found below (A1): there is hope that stability can be established, provided we adapt the policy on a slow time scale. This works well in experiments.

**Reorganization & Discussions (R#1 & R#4):** The high-level style was chosen to reach a broad audience, and motivate readers to read the supplementary material for technical details. Nevertheless, we welcome the reviewers criticisms: we will include more technical insights in a revision, making use of the additional page available for the final version. Parts of Section A.5.1 will be moved to the main body, highlighting the big challenge dealing with discontinuous dynamics. We will include the figure below as a navigational aid.

**Connection between Zap-Q and Newton-Raphson flow (R#1):** The Zap-SA algorithm is designed to approximate the Newton-Raphson flow: see lines 51-55, much like an Euler-discretization. Stochasticity presents a minor challenge. The complexity of the paper arises mainly because the vector field (7) is not continuous in applications to Q-learning.

**Experiments (R#1 & R#4):** We have conducted many more experiments over the past week in response to the reviewers. The figure below is the basis of the summary that follows

**1. Convergence of $\overline{f}(\theta_n)$ (R#4):** Fig A shows that $\|\overline{f}(\theta_n)\|$ converges to zero for the mountain car example.

**2. Fixing $\widehat{A}_n$ (R#1 R#3 & R#4):** We update $\widehat{A}_n$ every 50 samples in these experiments (see line 254 of the paper). In experiments this week, we found that performance degrades and is eventually unstable with increasing update interval. Fig B shows that the algorithm is not stable if the matrix is fixed, and the worst option is to take $\widehat{A}_n = A(\theta^*)$ for all $n$.

**3. Comparisons (R#1):** This week we tried the off-the-shelf implementation of DQN from bsuite (Osband et. al., ICLR, 2020, online). Using their default hyperparameters and the same environment setup used in our paper, we tested DQN with two layer NN, $100 \times 100$. The average rewards $\mathcal{R}(\phi^{\theta_n})$ estimated based on eq (31) are shown in Fig C. DQN with a big network is able to learn better policies, but it has high variance. Zap-Q is reliable and consistent in every example considered. The smooth behavior shown in Fig C is typical of Zap-Q, while DQN failed for the $16 \times 8$ NN.

**4. Comparison with GQ-learning (R#4):** GQ-learning [30] aims to solve (18) with linear function approximation. We applied this algorithm to mountain car using tile coding as basis vectors, but were unable to obtain meaningful results in this one week period. Fig D summarizes the behavior of the two algorithms for Baird's example [3].

**To R#3 on complexity of inverting $\widehat{A}_n$:** We are eager to explore more efficient techniques to approximate the Newton-Raphson flow. This may require advice from experts in the numerical analysis and optimization communities.

**To R#2 on the complexity of the paper:** We will add a short tutorial on SA and RL to the supplementary material.

[Meta-Review · NeurIPS 2020]

The reviewers are generally supportive of the paper. They have provided some very useful feedback, and I highly encourage the authors to incorporate that feedback. Primarily, it would be ideal to complete the paper reorganization as discussed, explain the limitations in the assumption on boundedness of the iterates, provide a toy example where the boundness assumption is not on its own enough to prevent divergence of Q-learning (i.e, even under that assumption, Q-learning diverges but Zap-Q does not) and finally to sweep over the parameters in the empirical comparison (even if that means the outcome is less positive for Zap-Q).